# Strong graphene oxide nanocomposites from aqueous hybrid liquid crystals

Maruti Hegde [1,2], Lin Yang [3], Francesco Vita [4], Ryan J. Fox [1], Renee van de Watering[2], Ben Norder [5], Ugo Lafont [6], Oriano Francescangeli[4], Louis A. Madsen [7], Stephen J. Picken[5], Edward T. Samulski[1] & Theo J. Dingemans [1,2✉]

Combining polymers with small amounts of stiff carbon-based nanofillers such as graphene or graphene oxide is expected to yield low-density nanocomposites with exceptional mechanical properties. However, such nanocomposites have remained elusive because of incompatibilities between fillers and polymers that are further compounded by processing difficulties. Here we report a water-based process to obtain highly reinforced nanocomposite films by simple mixing of two liquid crystalline solutions: a colloidal nematic phase comprised of graphene oxide platelets and a nematic phase formed by a rod-like high-performance aramid. Upon drying the resulting hybrid biaxial nematic phase, we obtain robust, structural nanocomposites reinforced with graphene oxide.

[1] Department of Applied Physical Sciences, University of North Carolina at Chapel Hill, Murray Hall, 121 South Road, Chapel Hill, NC 27599-3050, USA. [2] Faculty of Aerospace Engineering, Delft University of Technology, Kluyverweg 1, 2629 HS Delft, The Netherlands. [3] National Synchrotron Light Source II, Brookhaven National Laboratory, Upton, NY 11973, USA. [4] Dipartimento di Scienze e Ingegneria della Materia, dell'Ambiente ed Urbanistica and CNISM, Università Politecnica delle Marche, Via Brecce Bianche, 60131 Ancona, Italy. [5] Faculty of Applied Sciences, Delft University of Technology, Van der Maasweg 9, 2629 HZ Delft, The Netherlands. [6] European Space Technology and Research Centre, European Space Agency, Keplerlaan 1, 2201 AZ Noordwijk, The Netherlands. [7] Department of Chemistry and Macromolecules Innovation Institute, Virginia Tech, Blacksburg, VA 24061, USA. ✉email: tjd@unc.edu

Reinforcing polymers with nano-scale fillers like carbon nanotubes, graphene, and graphene oxide, is often touted as a prescription for fabricating low-density nanocomposites with exceptional mechanical properties[1,2]. However, property enhancement is critically dependent on stress transfer from the relatively soft polymer matrix to the stiff reinforcement[3]. The efficacy of stress transfer is, in turn, sensitive to the polymer morphology, the distribution of the filler particles, and ill-defined intermolecular interactions between the particle surface and polymer. To date a haphazard Edisonian approach to reinforcing inherently low-modulus polymers has been pursued by adding nano-scale fillers to melts or solutions of polymers with random coil secondary structures[4]. Consequently, strong nanocomposites remain elusive in part because of incompatibilities between amorphous polymer morphology and filler causing the latter to aggregate; composite fabrication is also impacted by processing difficulties[2]. Using a water-based process, we report that highly reinforced nanocomposite films can be obtained by simply mixing two uniaxial liquid crystalline solutions: (i) a colloidal nematic filler phase comprised of a suspension of aligned graphene oxide (GO) platelets and (ii) a nematic solution of rod-like high-performance aramid polymers. The resulting mixture is hybrid biaxial nematic phase[5]—a mesoscopic lyotropic nematic comprised of GO platelets embedded in a lyotropic polymeric nematic with respective directors orthogonal—with a locally stratified supramolecular organization. On drying, the stratified arrangement in the biaxial fluid is compressed into a uniplanar morphology yielding robust, structural nanocomposite films. The nanocomposite exceeds theoretical estimates—a 20 GPa modulus enhancement of the aramid and a strength enhancement of ~320 MPa without any decrease in the strain-at-break. The hybrid mesophase route employed in this work yields a uniform dispersion of filler and suggests a design strategy for fabricating structural nanocomposites. In short, we present a facile, robust route to overcome processing challenges and achieve well-dispersed nanocomposites that exhibit excellent load transfer between the matrix polymer and the GO reinforcing component.

## Results

**Hybrid biaxial nematics.** Poly(2,2′-disulfonyl-4,4′-benzidine terepthalamide) (PBDT; Fig. 1a) is an all-aromatic, rodlike polyelectrolyte related to the high performance aramid, Kevlar®[6]. PBDT forms a uniaxial nematic phase ($N^+$)[7–9] above 1.9 wt.% in water with its local director—the axis of alignment of quasi-parallel, high-persistence-length (rodlike) PBDT polymers, specified by $\mathbf{n}_P$; the $+$ sign indicates that the largest refractive index is parallel to $\mathbf{n}_P$. The PBDT solutions are biphasic between 1.9 and 12 wt.%, and fully nematic above the latter concentration (Supplementary Fig. 1). In the Onsager excluded-volume picture of lyotropic liquid crystal formation[10], a low onset concentration suggests a high aspect ratio. This results because the rod-like PBDT double helix aggregates into prolate particles—with an axial persistence length of ~260 nm (see Supplementary Note 1) and an aspect ratio of ~330[10–12]. Graphene oxide (GO), a precursor to "synthetic graphene"[13], has a lower modulus (~250 GPa)[14] than G, but the heterogeneity of the GO primary structure enables formation of a stable aqueous colloidal nematic phase above a critical concentration[15] with its director $\mathbf{n}_{GO}$ specifying the average direction of platelet normals; the largest refractive index is perpendicular to $\mathbf{n}_{GO}$, hence the negative sign in the phase designation, $N^-$. Micron-size GO platelets with (ideally) the thickness of a single carbon atom implies a very high aspect ratio (>$10^4$) and correspondingly a much lower liquid crystal onset concentration (~0.018 wt.%) and smaller biphasic range (0.018–0.9 wt.%)[15,16]; see Supplementary Figs. 1 and 2. The size selected GO platelets ≈3.6 μm in diameter

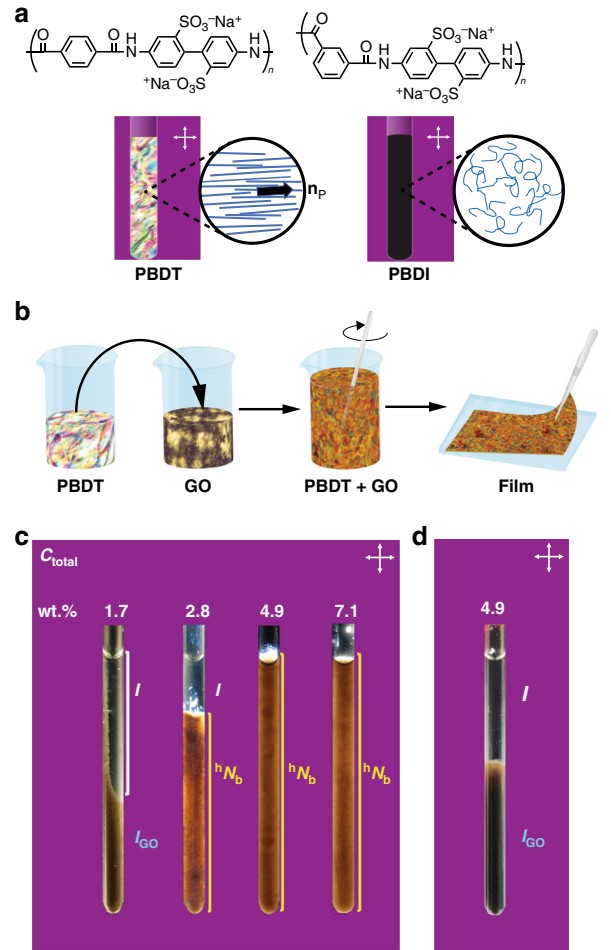

**Fig. 1 Molecular structures of polymers, nanocomposite synthesis and, hybrid biaxial nematics. a** Molecular structures of PBDT and PBDI, mesophases (images from crossed polarized optical microscopy have been cropped into NMR tube shapes using Adobe Illustrator™), and chain secondary structures (schematic insets). PBDT has a rectilinear primary structure and adopts a prolate, rod-like secondary structure that spontaneously forms a uniaxial nematic phase in water above ≈1.9 wt.%. The PBDT rods are locally aligned along a nematic director, $\mathbf{n}_P$; the solution appears bright between crossed polarizers. In contrast, the secondary structure of PBDI is a random coil globular shape with a primary structure that differs from that of PBDT only by the *meta*-linked isophthaloyl moiety; this substitution pattern introduces a bend into the PBDI secondary structure. PBDI solutions are isotropic and appear dark between crossed polarizers. **b** The water-based synthetic process to make PBDT + GO nanocomposite films. Stable PBDT + GO aqueous mixtures are obtained by simple mixing of the nematic components which on film casting and drying at 60 °C yields a nematic nanocomposite film (≈20 μm thick). **c** Phase stability of aqueous PBDT + GO mixtures. Nanocomposite precursor mixtures with a range of total solids concentrations ($C_{total}$ = 1.7, 2.8, 4.9, and 7.1 wt.%) each with a constant GO/(PBDT + GO) mass fraction ($F_{GO}$ = 0.0244). The 1.7 wt.% solution is isotropic, and the GO dispersion is metastable: Mild centrifugation (2 h at 600 RPM) results in the aggregation of flocculated GO in the lower isotropic phase ($I_{GO}$). Below $C_{total}$ = 3.6 wt.%, the solutions are biphasic, e.g., centrifugation of the $C_{total}$ = 2.8 wt.% mixture partitions into a PBDT-rich isotropic (I) upper phase coexisting with a birefringent PBDT + GO $^hN_b$ lower phase. Stable $^hN_b$ phases exist in nanocomposite precursor mixtures when $C_{total}$ > 3.6 wt.%. **d** Aqueous PBDI + GO solutions, irrespective of concentration, are isotropic and mild centrifugation partitions it into an upper PBDI rich isotropic phase and a lower $I_{GO}$ phase (e.g., $C_{total}$ = 4.9 wt.%). The backgrounds in **a**, **c**, **d** have been changed from black to pink for visualization purposes.

(Supplementary Fig. 3a), consist of single and bilayer GO according to TEM (Supplementary Fig. 3b), with a C/O ratio—indicative of functionalization degree—of 2.6 (Supplementary Fig. 3c), and in the $N^-$ phase are arranged with (undulating quasi-planar) GO surfaces locally parallel[17].

In mixtures of rods and plates, the competition between the excluded volumes of the two different shapes results in a concentration regime wherein the respective rod and plate directors $\mathbf{n}_P$ and $\mathbf{n}_{GO}$ are oriented along mutually perpendicular directions[18–20] to form a biaxial nematic phase ($N_b$)[21–23]. We anticipated that a hybrid liquid crystal phase would form when mesoscale GO platelets are dispersed in the $N^+$ phase of PBDT, analogous to the $^hN_b$ phases discovered by Mundoor et al. [5] In the latter, mesoscopic rodlike particles were added to a thermotropic molecular nematic, which adopted normal anchoring relative to the particle surfaces resulting in two orthogonal nematic directors. In our case tangential anchoring of PBDT on the GO surfaces maximizes the interactions between PBDT rods and GO platelets resulting in a hybrid phase with the two respective nematic directors $\mathbf{n}_P$ and $\mathbf{n}_{GO}$ orthogonal to one another. Hybrid PBDT + GO phases were prepared by mixing aqueous solutions of PBDT and GO (Fig. 1b and Methods) such that the mass fraction ($F_{GO}$) is held constant ($F_{GO} = GO/(PBDT + GO) = 0.0244$) in both the fluid mixtures (Supplementary Table 1)and in the solid films prepared by drying those mixtures. However, in order to create stable fluid dispersions of GO in solution it is essential to have the PBDT component in its liquid crystalline phase (Fig. 1c); when the total solids content, $C_{total}$ (the concentration of GO + PBDT), is less than 2.0 wt.%, the mixture is a simple isotropic solution and PBDT acts as a flocculant causing the GO to aggregate and sediment on mild centrifugation (Fig. 1c). In nematic hybrid mixtures i.e when $C_{total} > 2.0$ wt.%, GO exceeds its critical overlap concentration ($\emptyset^*_{GO}$) (see Supplementary Table 2 and associated calculations in Supplementary Information) resulting in orientational correlation between GO platelet normals. Furthermore, the effective volume per GO platelet ($V_{eff,GO}$) which is a measure of the accessible volume for GO platelets reduces below the corresponding GO overlap volume (Supplementary Table 2). Dispersions of GO in the $N^+$ phase of PBDT are stable even when the PBDT component is biphasic; for $C_{total} > 3.6$ wt.%, the entire

solution appears as a uniform stable $^hN_b$ phase. The detailed phase behavior of PBDT rods mixed with GO platelets is complex: For example, a 4.75 wt.% PBDT in water is biphasic ($I + N^+$), as is a 0.0120 wt.% GO suspension in water ($I + N^-$) (Supplementary Table 1). But the mixture of the two solutions ($C_{total} = 4.9$ wt.%) is a homogeneous $^hN_b$ phase that does not phase separate on centrifugation. The aspect ratio of the GO in mixtures is also a critical variable (Supplementary Figs. 3a and 4). Despite long-standing theoretical predictions[19,21], we are unaware of prior reports of stable rod + plate biaxial mesophases as de-mixing occurs spontaneously[24].

In contrast, the random coil polymer, poly(2,2′-disulfonyl-benzidine isophthalimide) (PBDI; Fig. 1a) does not form a liquid crystalline phase and merely acts as a polyelectrolyte flocculating agent for GO (Fig. 1c). As a result, contrasting the behavior of PBDI with the rodlike PBDT gives insights into how nanocomposite precursor phases impact mechanical properties.

Transmission small angle X-ray diffraction measurements were performed on the PBDT + GO mixtures with the geometries shown in Fig. 2a, at normal $\alpha = 90°$ and "edge-on" $\alpha \approx 15°$ incidence with respect to the containing cell surface.

At normal incidence, the scattering is essentially isotropic over the investigated concentration range (apart from the weak anisotropy from shear-induced orientation introduced during sample loading) (Supplementary Fig. 5a). In the edge-on geometry, the small-angle (low $\mathbf{q}$) GO-dominated scattering transforms from a circular pattern to an anisotropic azimuthal intensity distribution in the fully nematic mixtures (Fig. 2a and Supplementary Fig. 5a). The anisotropy appears to be generated from two sources: (i) shear-induced order on filling the cell with liquid crystalline fluids; (ii) anchoring preferences, i,e, the PBDT nematic adopts a homogeneous texture (random in-plane tangential anchoring) while the GO nematic favors homeotropic alignment (normal anchoring). The random alignment of $\mathbf{n}_P$ in the plane of the cell results in a two-dimensional mosaic structure of biaxial domains, which accounts for the isotropic scattering in the $^hN_b$ phase for $\alpha = 90°$. To reiterate, in the $^hN_b$ phases there is a preference for $\mathbf{n}_P$ to adopt tangential anchoring to surfaces (the cell and the GO) hence $\mathbf{n}_{GO}$ adopts orthogonal ordering (Fig. 2b). This interpretation is reinforced by the

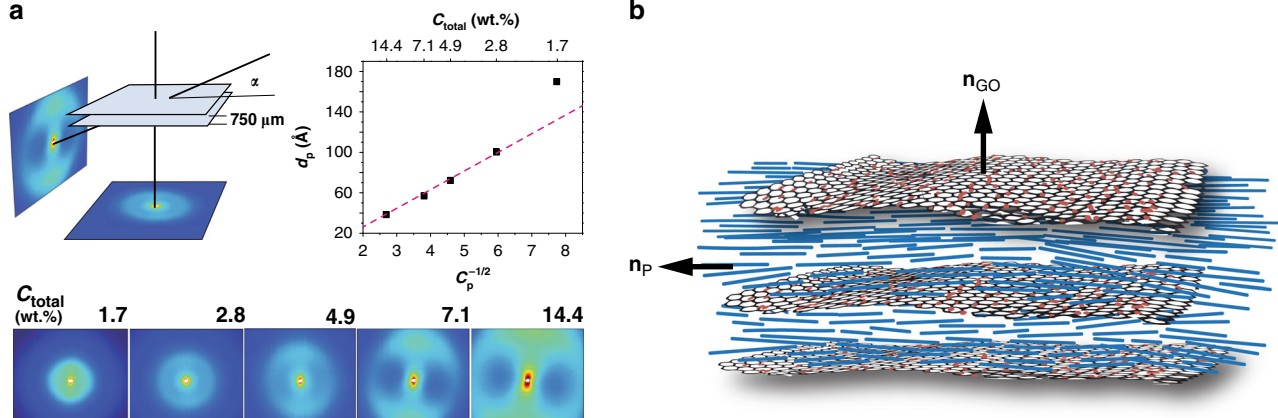

**Fig. 2 Characterization of PBDT + GO hybrid nematics. a** Small angle X-ray scattering from fluid PBDT + GO mixture contained in a transmission cell (parallel mica windows separated by a 750 μm spacer) for $\alpha \approx 15°$ ("edge-on") and $\alpha = 90°$ (normal incidence). In the hybrid phases the scattering from the PBDT is easily distinguished from scattering due to GO platelets; the former is at larger scattering angles[8] and the latter, dominated by GO scattering, is adjacent to the beam-stop. The non-uniform azimuthal intensity in the $\alpha \approx 15°$ scattering from the $^hN_b$ phase ($C_{total} = 4.9$, 7.1, and 14.4 wt.%) derives from a combination of flow-induced orientation on preparing the cell and preferential anchoring of both components of the $^hN_b$ phase. The inter-rod spacing $d_P$ for PBDT in the (biphasic) $^hN_b$ phase appears to be well behaved exhibiting a $d_P = C_P^{-1/2}$ scaling relationship; the unstable GO dispersion in the isotropic mixture ($C_{total} = 1.7$ wt.%) deviates from the scaling relationship. **b** A schematic diagram of the stratified supramolecular arrangement in a single monodomain of the $^hN_b$ phase wherein the directors $\mathbf{n}_{GO}$ and $\mathbf{n}_P$ are orthogonal.

angular dependence of the scattering (Supplementary Fig. 5a) where the inter-PBDT rod scattering and inter GO platelet scattering intensity is concentrated along the meridian for the edge-on diffraction patterns (Fig. 2a). The anisotropic, coaxial, meridional scattering from PBDT and GO along with the well behaved $d_P = C_P^{-1/2}$ scaling relationship[8] (Fig. 2a and Supplementary Fig. 5b) is evidence of a biaxial hybrid nematic showing the superposition of orthogonal PBDT rod and GO platelet directors, $\mathbf{n}_P$ and $\mathbf{n}_{GO}$ respectively (Fig. 2b). The stratified supramolecular arrangement of PBDT rods between GO platelets is reminiscent of the stratified morphology observed in a system comprising DNA rods adsorbed between 2D lipid membranes[25].

**Nanostructure of nanocomposite films.** Nanocomposite films with thickness ≈20 μm were prepared by a casting process (doctor blade, gap = 1 mm, casting velocity = 1 mm·s$^{-1}$) using thoroughly blended mixtures of separately prepared solutions of PBDT and colloidal GO in water. Polymer baseline properties are determined by preparing neat PBDT films; at some point during drying of neat (isotropic or biphasic) PBDT solutions, the critical concentration for lyotropic $N^+$ mesophase formation is exceeded. Since the $N^+$ phase adopts tangential anchoring on the flat casting substrate (untreated glass), $\mathbf{n}_P$ is parallel to the substrate interface, but the direction of $\mathbf{n}_P$ is not uniform throughout the fluid film (Supplementary Fig. 6 and Supplementary Fig. 7a). Unless the fluid is deliberately sheared while drying there is an absence of long-range order of $\mathbf{n}_P$. Typically, dimensionally stable, neat PBDT films exhibit a poly-domain mosaic morphology with the PBDT rods adopting a uniplanar orientation relative to the casting substrate. With the incident X-ray beam at $\alpha = 90°$ (normal incidence), neat PBDT films and all PBDT + GO nanocomposite film WAXS data associated with intramolecular and inter-molecular periodicities in PBDT (Supplementary Fig. 7a) have a nearly uniform intensity distribution, indicating a random arrangement of PBDT directors in the plane of the films (Supplementary Fig. 7a). The parallelism of $\mathbf{n}_P$ to the substrate i.e. the degree of uniplanar alignment of PBDT in the film can be evaluated by computing an order parameter $\langle P_2 = (3\cos^2\beta - 1)/2 \rangle$[26], where $\beta$ is the out-of-plane deviation of PBDT rods for edge-on incident beam data ($\alpha = 0°$, Fig. 3). The order parameter $\langle P_2 \rangle$ may be calculated from the azimuthal intensity distribution of the diffraction at $d_P = 3.8$ Å i.e. the inter-rod spacing in the dry films[26,27].

For neat PBDT films, $\langle P_2 \rangle \approx 0.6$ and is independent of the PBDT concentration in the casting solution. However, in the PBDT + GO mixtures, the $C_{total}$ has a pronounced influence on PBDT organization within the nanocomposite films. Casting and drying from stable, hybrid nematic phases (e.g., $C_{total} > 4.9$ wt.%) results in stratified supramolecular organization i.e. narrower distributions of the inter-rod scattering intensity (higher in-plane alignment) with a higher $\langle P_2 \rangle$ value (>0.8). By contrast, diffuse inter-rod (Fig. 3 and Supplementary Fig. 7a) scattering patterns with lower $\langle P_2 \rangle$ values are obtained in composite films cast from the metastable PBDT + GO mixtures (e.g., $\langle P_2 \rangle \approx 0.25$ and $\approx 0.4$ for $C_{total} = 1.7$ and 2.9 wt.%, respectively). In nanocomposite films cast from $^hN_b$ mixtures, the meridional scattering for the nanocomposite SAXS measurements (Fig. 3 and Supplementary Fig. 7a) and the well-defined isotropic scattering at $\alpha = 90°$ suggests the presence of relatively well-defined aggregates (PBDT fibrils)[7] separating GO platelets with high in-plane orientational order ($\langle P_2 \rangle > 0.9$) (Supplementary Fig. 7b). We do note that the prohibitively high viscosity of $C_{total} = 14.4$ wt.% prevents the preparation of uniform nanocomposite films using a doctor blade. PBDI + GO nanocomposite films are brittle with isotropic

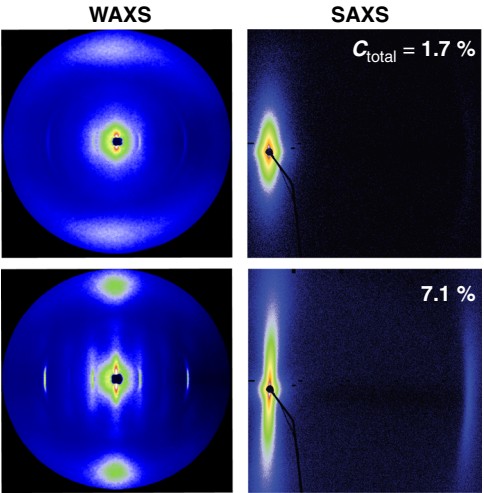

**Fig. 3 Nanostructure of PBDT + GO nanocomposite film.** WAXS and SAXS scattering results measured at $\alpha = 0°$ (film edge-on scattering) for PBDT + GO nanocomposite films cast from $C_{total} = 1.7$ wt.% ($I$ phase) and 7.1 wt.% ($^hN_b$ phase). In WAXS patterns, distinct intramolecular and intermolecular periodicities in PBDT are only observed in nanocomposite films prepared from fully monophasic $^hN_b$ mixture. The stronger azimuthal dependence of PBDT scattering in films cast from $C_{total} = 7.1$ wt.% indicates uniplanar distribution of PBDT rods i.e. a collapsed version of the $^hN_b$ phase wherein the directors are orthogonal with $\mathbf{n}_P$ in the film plane and $\mathbf{n}_{GO}$ normal to the film giving rise to the stratified morphology. In these nanocomposites, the highly anisotropic meridional scattering from SAXS measurements indicates PBDT aggregates and GO platelets have in-plane orientation.

scattering exhibiting no significant change in PBDI morphology by adding the GO filler (Supplementary Fig. 8).

**(Thermo) Mechanical properties.** The casting solution concentration $C_{total}$, which controls the perfection/homogeneity of the hybrid mesophase, strongly affects the nanocomposite stiffness determined from storage modulus ($E'$) values using dynamic mechanical thermal analysis (DMTA) (Supplementary Fig. 9). Pure PBDT films have $E'$ values of 10 GPa, and for this high-performance polymer, $E'$ is invariant with temperature up to 400 °C (Supplementary Fig. 9). The magnitude of the mechanical enhancement of PBDT with GO depends on the phase of the casting solution—isotropic or (hybrid) liquid crystalline—which in turn affects the dispersion quality (Fig. 4a and Supplementary Fig. 10).

Films cast from the unstable isotropic or biphasic hybrid mixtures (e.g., $C_{total} = 1.7$ wt.% ($I$) or 2.8 wt.% ($I + {^hN_b}$)) exhibit visible GO aggregation on a length scale of several hundred of microns, (Supplementary Fig. 10). Additionally, the low in-plane alignment of the polymer ($\langle P_2 \rangle \approx 0.25$) in films prepared from the isotropic hybrid mixture ($C_{total} = 1.7$ wt.%) results in nanocomposite $E'_{NC}$ values marginally lower ($E'_{NC} = 8.3$ GPa $< E'_{PM} = 10$ GPa) than that of pure PBDT. Nanocomposite films derived from the biphasic hybrid mixture ($C_{total} = 2.8$ wt.%) exhibit a modest $E'$ enhancement ($\Delta E = E'_{NC} - E'_{PM} = 1.7$ GPa) despite GO aggregation (Supplementary Fig. 10a) and polymer alignment ($\langle P_2 \rangle \approx 0.4$) lower than that of neat PBDT ($\langle P_2 \rangle \approx 0.6$). This observed $\Delta E$ lies within the range of values commonly obtained for commodity plastics modified with GO (Fig. 4a) and indicates that the contribution is primarily from stress-transfer to the mechanical reinforcement. The stratified morphology of nanocomposite films cast from monophasic $^hN_b$ mixtures ($C_{total} = 4.9$ and 7.1 wt.%) results in large $E'$ values of 25.5 and 33.2 GPa i.e. $\Delta E = 15.5$ GPa and 23.2 GPa respectively that are greater than all

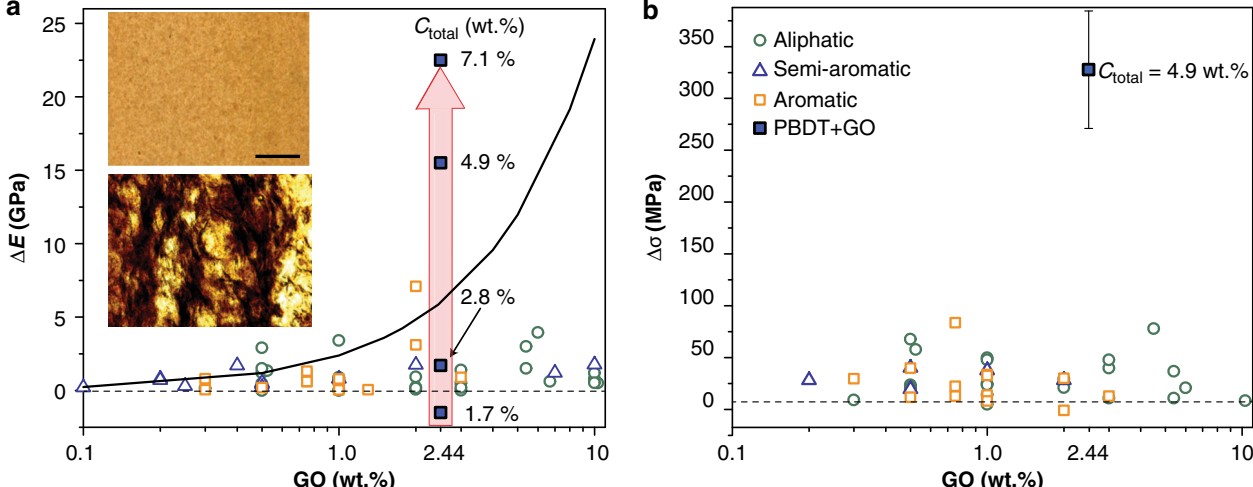

**Fig. 4 Mechanical properties of PBDT + GO nanocomposites. a** The enhancement in storage modulus ($E'$) from dynamic mechanical thermal analysis for PBDT + GO nanocomposites (denoted by filled blue squares). The differences between the maximum film modulus of GO nanocomposites ($E_{NC}$) and the corresponding values of polymer matrix modulus ($E_{PM}$), $\Delta E$, is tabulated for literature data for different classes of polymers (aliphatic, semi-aromatic, and aromatic denoted by open-turquoise circles, open blue triangles and orange squares respectively) reported for either dynamic thermo-mechanical analysis ($E'$) or tensile measurements ($E$) of films. Theoretical estimate for $\Delta E$ was calculated from the rule of mixtures[3] using densities of 1.4 and 2.0 g.cm$^{-3}$ for PBDT and GO respectively, $E_{GO} = 250$ GPa, GO aspect ratio = 3500 and perfect in-plane orientation within the nanocomposite; $\Delta E$ is invariant for $E_{PM}$ values in the 0.1–10 GPa. For PBDT + GO nanocomposites, the $\Delta E$ strongly depends on $C_{total}$ of the hybrid film casting mixture; $\Delta E = 23$ GPa is obtained for PBDT + GO films when cast from $C_{total} = 7.1$ wt.%. Optical microscopy images of nanocomposites also indicate that GO dispersion quality is also dependent on $C_{total}$. The scale bar (200 µm) is the same for both optical microscopy images. **b** Strength enhancement from tensile stress-stress measurements. The Y-error bar represents the standard deviation from the average. The enhancement $\Delta\sigma = (\sigma_{NC} - \sigma_{PM})$ is the difference between the nanocomposite's maximum tensile strength ($\sigma_{NC}$) and that of the polymer matrix ($\sigma_{PM}$); while $\sigma_{NC}$ is dependent on the GO loading, $\Delta\sigma$ does not show any significant correlation with GO content. Reported $\Delta\sigma$ values for GO-polymer nanocomposites are indicated for optimum GO loadings for a variety of polymer matrices (see Supplementary Data 1); for PBDT + GO nanocomposite cast from $C_{total} = 4.9$ wt.% $^h N_b$ mixture, $\Delta\sigma = 320$ MPa.

previously reported glassy polymer + GO nanocomposites (Fig. 4 and Supplementary Data 1). The simple fabrication of the nanocomposite films described herein yields $\Delta E$ values that are comparable to polymer + GO nanocomposite fibers[4,28,29]. The lack of modulus enhancement in nanocomposites cast from unstable PBDT + (small GO) or the isotropic PBDI + GO precursor mixtures ($C_{total} = 4.9$ wt.% in both) (Supplementary Fig. 9) further highlights the importance of preparing nanocomposites from stable, hybrid biaxial nematic mixtures.

The stress-strain analysis of nanocomposite films having a "compressed" uniplanar $^h N_b$ morphology exhibits enhanced mechanical properties (e.g., Young's modulus, tensile strength, and strain-at-break) relative to the uniplanar morphology of pure PBDT films (Supplementary Fig. 10c, $C_{total} = 4.9$ wt.% and Supplementary Table 3). We observe an tensile strength enhancement of 320 MPa (Fig. 4b) without any decrease in the strain at break—contrary to the commonly reported propensity for brittle fracture that accompanies an increase in nanocomposite stiffness[4]. A comparison of the largest (and the average) strain-at-break values for PBDT and PBDT + GO (Supplementary Table 3) indicates that the strain-at-break is improved in the nanocomposite films. Generally, mechanical properties, such as modulus, strength, and strain at break of rigid-rod polymers such as Kevlar® increase with polymer orientation[30]. In PBDT + GO nanocomposite films, the enhancements in modulus, strength, and strain at break result from the increased PBDT + GO orientation; this is in turn derived from the unique supramolecular organization of the rodlike polymer and platelike filler—a stable, stratified morphology obtained from drying the hybrid biaxial nematic solutions.

## Discussion
The striking enhancements in modulus and strength for nanocomposites prepared from hybrid biaxial nematic mixtures clearly

demonstrates that rigid-rod high-performance polymers are excellent matrix materials for GO-based nanocomposites. The phase stability of the aqueous mixtures provides a variable for control of the morphological organization in GO-reinforced films. Our observations provide design and optimization strategies for preparing nanocomposite materials from anisotropic precursors, thus enabling polymer nanocomposites with properties that have remained elusive despite decades of effort. Blending liquid crystalline phases of mesoscale reinforcements with liquid crystalline phases of high-performance polymers may provide a pathway for overcoming the putative Achilles heel of structural nanocomposites—processing a wide range of filler loadings to generate large mechanical enhancements.

## Methods
**Materials**. The monomers terephthaloyl chloride (TC) and isophthaloyl chloride (IPC) were purchased from Sigma Aldrich. Only freshly sublimed terephthaloyl chloride and isophthaloyl chloride were utilized for the polymerization reactions. The diamine monomer, 4,4′-diaminobiphenyl-2,2′-disulfonic acid hydrate (95%) (BDSA) was purchased from Alfa Aesar and purified before use. Polyethylene glycol 300 (PEG-300) was purchased form Sigma Aldrich. Natural graphite flakes from Sigma Aldrich were used to make graphene oxide. SpectraPor 1 dialysis membranes were purchased from Sigma Aldrich.

**Methods**. *Polymer synthesis.* The PBDT and PBDI polymers with Na$^+$ counter ions were synthesized according to the interfacial procedure reported by Sarkar et al.[6] PBDT synthesis is as follows: A 2 L three-neck round-bottom flask equipped with a mechanical stirrer was charged with sodium carbonate 1.59 g (15 mmol), dry BDSA (2.58 g, 7.5 mmol), PEG 300 (2.4 g) and 500 mL deionized water. This mixture was stirred at 1000 RPM for ~15 min. After 15 min, TPC (1.52 g, 7.5 mmol) dissolved in 100 mL of chloroform was added to the reaction mixture and polymerized for 15 min. A rotary evaporator enabled removal of chloroform from the mixture and the polymer was obtained by precipitation of the aqueous solution in 2 L of acetone. The precipitate thus collected was redissolved in ~200 mL water and precipitated and filtered from 2 L of acetone. This procedure was repeated thrice—until pH of the aqueous solution was 7. PBDI was synthesized

using a similar procedure; instead of TPC, IPC dissolved in dichloromethane was used as the diacid chloride solution. Both polymers were dried at 150 °C for 1 h in a vacuum oven prior to use. The measured inherent viscosity of PBDT and PBDI in water are ~30 dL·g$^{-1}$ and ~4 dL·g$^{-1}$—these values are consistent with previously reported values[9].

*Synthesis of graphene oxide.* Graphite was oxidized using the Hummers method[31]. The work-up procedure to obtain liquid crystalline graphene oxide solution is as follows: The Hummers method yields a thick slurry that contains a mixture of graphene oxide, graphite oxide, unreacted graphite and salts of potassium and manganese. The slurry was filtered over a Buchner funnel and the filtrate collected. The filtrate was centrifuged at 500 RPM for 1 h to remove any residual graphitic impurities. The supernatant was washed with 200 mL water (1×), 300 mL of 30% HCl (1×) and with 200 mL ethanol (2×). After each washing step, the solution was subjected to 500 RPM centrifugation for 2 h and the supernatant collected. After all the washing steps, the resultant 1 L solution was subjected to 1.5 h of low-intensity bath sonication and the mixture centrifuged at 500 RPM for 30 min. The yellow-brown supernatant was separated from the sediment and collected in a beaker. The sediment from the centrifugation was added to water (≈50 mL) and bath sonicated for 1.5 h followed by subsequent centrifugation at 500 RPM for 30 min. This step was repeated until a clear supernatant was obtained. A total of ≈1 L supernatant was collected in this manner. The solution was dialyzed for 4 days using SpectraPor 1 Dialysis membranes to remove residual ionic impurities. The dialyzed solutions were subjected to 3500 RPM centrifugation for 1.5 h to fractionate the graphene oxide into isotropic and liquid crystalline fractions. The concentration was determined from thermo-gravimetric and UV-Vis spectroscopy measurements.

*Polymer + GO nanocomposite synthesis.* A typical hybrid mixture synthesis, for e.g. PBDT + GO with $C_{total}$ = 4.9 wt.% and the nanocomposite film is as follows: 300 mg of pre-dried PDBT polymer is dissolved in 3 mL deionized water. This is added dropwise (over ≈15 min) to a bottle containing 3 mL 0.25 wt.% LC solution of GO while continuously mixing to homogenize the mixture. The solution was doctor bladed onto an untreated glass plate (gap = 1 mm, velocity ≈ 1 mm·s$^{-1}$) and placed in an oven at 60 °C overnight resulting in a nanocomposite film. The film was removed from the glass plate by immersing the plate in an acetone bath.

*A High-resolution JEOL scanning electron microscope (HR-SEM)* was utilized to measure GO flake dimensions and study the cross-sections of the films. For statistical analysis of the GO flake dimensions, the graphene oxide flakes were first deposited on a Si wafer using a rudimentary Langmuir Blodgett approach based on the work by Cote et al.[32] Graphene oxide samples were mixed with methanol in a 1:5 (water/methanol) ratio. This was done to obtain well-dispersed GO flakes at the air-water interface. This solution was carefully pipetted on to a water trough. A Si wafer was dipped into this solution by immersing it carefully and later dried in a vacuum oven at 80 °C overnight. This enabled us to avoid imaging problems due to deposition of the GO on top of each other that makes dimension analysis difficult. The operating conditions of the SEM are as follows: working distance = 8 mm, operating voltage 1 kV, probe current = 20 μA, and imaging mode utilized was the lower secondary electron image.

*Transmission electron microscopy (TEM)* imaging was performed using a FEI Tecnai TF20 electron microscope operating at 200 kV. The GO aqueous solution was diluted by a factor of 100. The diluted samples were deposited on to a Quantifoil holey carbon grid with Cu-200 mesh using a pipette. The carbon grids were dried in air at 25 °C for ≈30 min.

*X-ray Photoelectron spectroscopy (XPS)* measurements were performed using a Kratos Axis Ultra DLD X-ray Photoelectron Spectrometer. The measurements were performed at room temperature and at a chamber pressure of 10$^{-6}$ mbar. The GO was deposited onto Au@Si substrates prior to analysis. The binding energy reported are within ±0.1 eV. The C/O ratio of GO was measured by dividing the atom percent of carbon by the atom percent of oxygen.

*The nematic phase fraction (%)* of PBDT solutions and PBDT + GO hybrid mixtures were analyzed by filling 5 mm NMR tubes and centrifuging them for 2 h at 600 RPM. For GO solutions, the filled NMR tubes were centrifuged at 8000 RPM for 2 h—similar results are achieved by leaving the tubes undisturbed at ambient conditions for ≈3.5 months. The nematic phase fraction (%) is calculated by dividing the height of the nematic phase by the total solution height. The centrifuged NMR tubes were placed between crossed polarizers on a white light background and imaged using an iPhone 8 camera.

*A Leica DM-LM optical microscope* equipped with crossed polarizers was used to image liquid crystalline samples. The samples were first transported in to a rectangular Vitrotrube® capillary (I.D. = 0.2 mm) by capillary action.

*Transmission small angle X-ray scattering* measurements on hybrid mixtures were carried out at the Life Science X-ray Scattering (LiX) beamline of NSLS-II. The X-ray energy was 13.5 keV (0.918 Å). The sample-to-detector distance was 3.23 m. The sample holder consisted of two mica plates (~20 micron thick) separated by a 0.75 mm acrylic spacer with a central aperture of 4.6 mm diameter. One of the mica plates was first glued to the spacer. The second mica plate was then placed onto the other side of the spacer (like a microscope cover slip), after the sample was pipetted into the aperture in the spacer. The angle between the incident X-ray beam and the mica plates was set at either α ≈ 15° (edge-on) or α = 90° (normal incidence) to analyze the orientation of PBDT + GO composite precursor solutions. In order to ensure that the beam cleared the spacer when the sample was

rotated (up to 15°), the X-ray beam was focused to a spot size of ≈5 micron. The detector gaps were filled using centro-symmetry.

*X-ray scattering studies* on nanocomposite films were performed at room temperature both in-house, using a Bruker AXS D8 Discover diffractometer in transmission mode with a CuKα-radiation source, and at the BM26B DUBBLE beamline of the European Synchrotron Radiation Facility (ESRF), Grenoble, France. For every nanocomposite, four layers of the thin films were mounted on a support with the film surface either orthogonal to the beam direction, or nearly parallel (with an offset of ≈1°). In-house WAXS measurements were performed using a distance of 6 cm between the sample and the detector and an exposure time of 5 min. In-house SAXS was performed by placing the samples at a distance of 30 cm from the detector for time period of 10 min. Synchrotron measurements of composite films were performed using a beam wavelength of 0.827 Å/1.033 Å and a sample-to-detector distance of 0.173 m/2.468 m for WAXS/SAXS, respectively. A vacuum chamber was placed between the sample and detector in SAXS measurements to reduce the scattering due to air. For 2D WAXS synchotron patterns, the experimental semi-circular 2D WAXS patterns were mirrored along the equator for easier visualization.

*Stress-strain analysis* were performed using a RSA-G2 Solids Analyzer (TA Instruments) with a 32 N load cell. Free-standing films were cut into rectangular strips of approximate cross-sectional area 1.5 × 0.03 mm$^2$ (width × thickness). A 10 mm gauge length at a constant linear displacement rate of 0.1 mm·min$^{-1}$ was used for tensile measurements. The materials were found to be strain rate independent within the range of 0.01–1 mm·min$^{-1}$ linear displacement rate. Specimens were tested under dry conditions at 25 °C under a nitrogen atmosphere. Specimens for dry tensile testing were treated at 200 °C for 20 min under nitrogen in the RSA-G2 forced convection oven to eliminate water. The samples were equilibrated at 25 °C under nitrogen before tensile measurements.

*Dynamic mechanical thermal analysis (DMTA)* was performed with a Perkin-Elmer Diamond DMTA. DMTA experiments on the films were performed at a frequency of 1 Hz at a heating rate of 2.0 °C/min using films having approximate dimensions of 20 × 3 × 0.020 mm.

## Data availability

All relevant data generated during and/or analyzed during the current study are available from the corresponding author upon reasonable request. No data in this paper are restricted in terms of availability. The source data underlying Figs. 2a, 4a and Supplementary Figs. 1, 2c, 3a, c, 5b, 7b, 9, and 10c are provided as a Source Data file.

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

## Acknowledgements

We acknowledge Jianwei Gao's (Delft University of Technology) help in the synthesis of PBDT and PBDI. We thank Jure Zlopasa (Delft University of Technology) for helpful discussion on nanocomposites and Ying Wang (Virginia Tech) for helpful discussions on X-ray scattering of PBDT. We also acknowledge M. Pisani (Università Politecnica delle Marche), D. Hermida-Merino (Netherlands Organization for Scientific Research, DUBBLE at ESRF) and C. Ferrero (ESRF) for support in the synchrotron XRD measurements. T.J.D. acknowledges his N.W.O. VIDI grant, project no. 07560, which supported a major part of this work. L.A.M. acknowledges the U.S. National Science Foundation under award number DMR 1810194. The LiX beamline is part of the Life Science Biomedical Technology Research resource, jointly supported by the National Institute of Health, National Institute of General Medical Sciences under Grant P41 GM111244, and by the Department of Energy Office of Biological and Environmental Research under Grant KP1605010, with additional support from NIH Grant S10 OD012331. NSLS-II is a U.S. Department of Energy (DOE) Office of Science User Facility operated for the DOE Office of Science by Brookhaven National Laboratory under Contract No. DE-SC0012704.

## Author contributions

M.H. contributed to conceptualization, synthesis, DMTA, SEM, in-house XRD, L.C. characterization, and composing/revising the paper. L.Y. performed and analyzed scattering of hybrid mixtures and revised paper. F.V. and O.F. performed and analyzed X-ray measurements of films and contributed to paper revision. R.v.d.W. contributed to synthesis and SEM measurements. R.J.F. contributed to synthesis, tensile measurements, and paper revisions. U.L. performed TEM measurements. B.N. and S.J.P. performed in-house X-ray measurements and contributed to paper revisions. L.A.M. contributed to paper revisions. S.J.P. and L.A.M. also aided in conceptualization. E.T.S. and T.J.D. contributed to conceptualization, data analysis and composing/revising the paper. T.J.D. secured funding and supervised the project.

## Competing interests

The authors declare no competing interests.
