## [Peer Review File · Nature Communications]

Reviewers' comments:

Reviewer #1 (Remarks to the Author):

The manuscript presents a breakthrough that is worthy of publication in Nature Communications: mixing a liquid crystalline solution of meso-scale reinforcements with a liquid-crystalline solution of a high-performance polymer overcomes the long-standing obstacles to producing nanocomposite structural materials. I believe this result would interest a broad range of readers interested in materials chemistry, liquid crystals and nanoscience in general. The result was discovered quite deliberately: the narrow concentration range in which they demonstrate facile, stable dispersion of graphene oxide (GO) in a particular high-performance polymer (PBDT) would be difficult to discover if not for the insight that nematic order in both of the individual solutions is essential.

By choosing a concentration of graphene oxide that is in the nematic phase and by choosing a stiff matrix polymer at a concentration in its nematic phase, simply mixing the two solutions provides a hybrid nematic single-phase. Casting a film from this homogeneous liquid-crystalline solution provides a scalable route to nanocomposites in which a large increase in stiffness and strength relative to the matrix polymer is actually realized. The evidence provided supports the assertion that casting from a single-phase hybrid nematic solution is critical: The usual difficulty in dispersing GO into a polymer solution is observed when the solution is in the isotropic phase or is biphasic. Characterization of the resulting nanocomposite shows that the liquid crystalline order of the GO solution persists in the solid state (GO sheets parallel to one another), creating a stratified morphology of GO alternating with PBDT. Their approach provides good adhesion between the GO and PBDT: with 2.4%wt GO, both stiffness and tensile strength more than double relative to PBDT (which has elastic modulus 10GPa up to 400°C and tensile strength ca. 200MPa). The SI Table is very helpful, enabling the reader to confirm that their materials provide unprecedented tensile strength.

Major revisions are required prior to publication:

1. The present material is NOT biaxial. "biaxial" materials, including liquid crystals, have three distinct optical axes (e.g., cited ref 6= Munder, H; Park, S; Senyuk, B., Wensink, H. H., Smalyukh, I. I. "Hybrid molecular-colloidal liquid crystals." *Science*: 360, 768-771 (2018)). The scattering patterns observed with the x-ray beam normal to the GO are isotropic. Therefore, the present material has only two optical axes (all directions perpendicular to the normal to the GO are indistinguishable).
2. Quantitative information on the persistence length and the overall length of PBDT must be provided.
3. Due to the potential importance of electrostatic interactions in the assembly process, the authors should provide the carboxylic acid functionality of the GO, e.g., T. Szabo, E. Tombacz, E. Illes and I. Dekany, *Carbon*: v. 44, pp. 537-545 (2006).
4. The paper must cite and describe "Structure of DNA-cationic liposome complexes: DNA intercalation in multilamellar membranes in distinct interhelical packing regimes," Radler, JO; Koltover, I; Salditt, T; Safinya, CR; *Science*: v. 275, pp. 810-814 (1997) DOI: 10.1126/science.275.5301.810. Abstract: Cationic liposomes complexed with DNA (CL-DNA) are promising synthetically based nonviral carriers of DNA vectors for gene therapy. The solution structure of CL-DNA complexes was probed on length scales from subnanometer to micrometer by synchrotron x-ray diffraction and optical microscopy. The addition of either linear lambda-phage or plasmid DNA to CLs resulted in an unexpected topological transition from liposomes to optically birefringent liquid-crystalline condensed globules. X-ray diffraction of the globules revealed a novel multilamellar structure with alternating lipid bilayer and DNA monolayers. The lambda-DNA chains form a one-dimensional lattice with distinct interhelical packing regimes. Remarkably, in the isoelectric point regime, the lambda-DNA interaxial spacing expands between 24.5 and 57.1 angstroms upon lipid dilution and is indicative of a long-range electrostatic-induced repulsion that is possibly enhanced by chain undulations.
5. If the authors already have examined the effects of ionic strength, counterion type and/or pH, it

would enhance the present publication to mention those results (even if detailed examination of these effects is beyond the scope of this communication).

6. In relation to strain at break, the data show that the strain at break did not decrease (sufficient to make the point that an increase in stiffness was achieved without reducing strain at break); however, the data are not sufficient to state that the strain at break increased by 40% in PBDT+GO relative to PBDT. ED Fig 10c reveals a large standard deviation in the strain at break for the GO nanocomposites (ca. 0.34) which, combined with the small number of replicates (5), precludes a precise determination of the strain at break for the PBDT+GO nanocomposites. I was unable to find strain-at-break values in the body or the ED; using the data in ED Fig 10c to estimate the mean strain at break (1.26% for PBDT and 1.49% for PBDT+GO) yields a difference less than 40% and a significant probability (ca. 20%) that the PBDT+GO has the same strain at break as PBDT.

7. The presentation could be greatly improved. For example, the opening repeats often stated but rarely realized statement that "Reinforcing polymers with nano-scale fillers, such as carbon nanotubes or graphene oxide, is a prescription for fabricating low-density nanocomposites with exceptional mechanical properties," which is contradicted by the cited references: ref 2 states "after nearly two decades of work in the area, questions remain about the practical impact of nanotube and graphene composites. This uncertainty stems from factors that include poor load transfer, interfacial engineering, dispersion, and viscosity-related issues that lead to processing challenges in such nanocomposites." Indeed, what makes this manuscript exciting is that it provides a facile, robust route to overcome processing challenges and achieve well-dispersed GO and excellent load transfer between the matrix polymer and the GO—which has proven elusive despite decades of effort.

Reviewer #2 (Remarks to the Author):

This is an interesting paper upon the preparation and mechanical properties of a polymer-based nanocomposite reinforced by graphene oxide (GO). The authors show that considerable modulus improvement can be realized through confining the GO in a high-modulus aligned liquid crystalline polymer, consistent with the latest theories of nanocomposite reinforcement that have been well cited in the text. I have to raise a couple of issue that need further information or comment.

1) The authors show that their findings are consistent with the rule of mixtures, although as far as I can tell they only quote weight fractions of GO. Volume fractions are used in the rule of mixtures so these should be quoted, e.g. for the 7.1 wt% sample, and/or densities given so that the reader can do the conversion for themselves.

2) A number of sophisticated analytical techniques have been employed. It is notable that Raman spectroscopy, which would have been very useful for their materials (e.g. to characterize orientation or monitor stress transfer), has not been used. The authors should explain why (e.g. problems with fluorescence?) and discuss how it might have helped.

Robert J Young

Rebuttal for

Strong Graphene Oxide Nanocomposites from Aqueous Hybrid Liquid Crystals

Authors: Maruti Hegde^{1,7}, Lin Yang², Francesco Vita³, Ryan J. Fox⁷, Renee van de Watering¹, Ben Norder⁴, Ugo Lafont⁵, Oriano Francescangeli³, Louis A. Madsen⁶, Stephen J. Picken⁴, Edward T. Samulski⁷, Theo J. Dingemans^{1,7*}

Affiliations:

¹Faculty of Aerospace Engineering, Delft University of Technology Kluyverweg 1, 2629 HS Delft, The Netherlands.

²National Synchrotron Light Source II, Brookhaven National Laboratory, Upton, NY 11973, United States of America.

³Dipartimento di Scienze e Ingegneria della Materia, dell'Ambiente ed Urbanistica and CNISM, Università Politecnica della Marche, Via Breccie Bianche, 60131 Ancona, Italy.

⁴Faculty of Applied Sciences, Delft University of Technology, Van der Maasweg 9, 2629 HZ, Delft, The Netherlands.

⁵Components Technology & Space Materials Division, European Space Agency, Keplerlaan 1, 2201 AZ Noordwijk, The Netherlands.

⁶Department of Chemistry and Macromolecules Innovation Institute, Virginia Tech, Blacksburg, VA 24061, United States of America.

⁷Department of Applied Physical Sciences, University of North Carolina at Chapel Hill, Murray Hall, 121 South Road, Chapel Hill, NC 27599-3050, United States of America.

*email: tjd@unc.edu

Notes: Reviewers' comments and text are shown in **black**, while author responses and new text are shown in **blue**.

In the manuscript PDF, edits made based on reviewer suggestions are highlighted in **green**. All edits made to meet *Nature Communication* formatting requirements are highlighted in **pink**.

Reviewer's comments:

Reviewer #1

Remarks to the Author:

A. The manuscript presents a breakthrough that is worthy of publication in Nature Communications: mixing a liquid crystalline solution of meso-scale reinforcements with a liquid-crystalline solution of a high-performance polymer overcomes the long-standing obstacles to producing nanocomposite structural materials. I believe this result would interest a broad range of readers interested in materials chemistry, liquid crystals and nanoscience in general. The result was discovered quite deliberately: the narrow concentration range in which they demonstrate facile, stable dispersion of graphene oxide (GO) in a particular high-performance polymer (PBDT) would be difficult to discover if not for the insight that nematic order in both of the individual solutions is essential. By choosing a concentration of graphene oxide that is in the nematic phase and by choosing a stiff matrix polymer at a concentration in its nematic phase, simply mixing the two solutions provides a hybrid nematic single-phase. Casting a film from this homogeneous liquid-crystalline solution provides a scalable route to nanocomposites in which a large increase in stiffness and strength relative to the matrix polymer is actually realized. The evidence provided supports the assertion that casting from a single-phase hybrid nematic solution is critical: The usual difficulty in dispersing GO into a polymer solution is observed when the solution is in the isotropic phase or is biphasic. Characterization of the resulting nanocomposite shows that the liquid crystalline order of the GO solution persists in the solid state (GO sheets parallel to one another), creating a stratified morphology of GO alternating with PBDT. Their approach provides good adhesion between the GO and PBDT: with 2.4%wt GO, both stiffness and tensile strength more than double relative to PBDT (which has elastic modulus 10GPa up to 400°C and tensile strength ca. 200MPa). The SI Table is very helpful, enabling the reader to confirm that their materials provide unprecedented tensile strength.

We thank the referee for these positive comments. We believe that the combination of facile processing and impressive mechanical performance makes this work relevant for a wide range of potential applications. Moreover, we believe this work highlights the role of the polymer structure, which is often not considered by the nanocomposite community.

Major revisions are required prior to publication:

1. The present material is NOT biaxial. “biaxial” materials, including liquid crystals, have three distinct optical axes (e.g., cited ref 6= Munder, H.; Park, S.; Senyuk, B.; Wensink, H. H.; Smalyukh, I. I. “Hybrid molecular-colloidal liquid crystals.” *Science*: 360, 768-771 (2018)). The scattering patterns observed with the x-ray beam normal to the GO are isotropic. Therefore, the present material has only two optical axes (all directions perpendicular to the normal to the GO are indistinguishable).

While we agree with the reviewer that the hybrid LC materials are not uniformly-oriented biaxial phases, our composites are considered locally biaxial materials, see Madsen, L. A., Dingemans, T. J., Nakata, M., Samulski, E. T. Thermotropic Biaxial Liquid Crystals. *Phys. Rev. Lett.*, **92**, 145505-1–145505-4. Kouwer, P. H., Mehl, P. H. Full Miscibility of Disk- and Rod-Shaped Mesogens in the Nematic Phase. *J. Am. Chem. Soc.*, **125**, 11172-11173 (2003). (Ref 21) Luckhurst, G. R., Sluckin, T. J.: Biaxial Nematic Liquid Crystals: Theory, Simulation and Experiment (John Wiley & Sons, Southampton, 2015). To be more specific: Observing three distinct optical axes would be the case for an oriented mono-domain biaxial LC – as shown by Munder *et al.* (ref. 6). The PBDT+GO nematic hybrid mixture samples we report in this article are *locally* biaxial. The local uniaxial director of PBDT is tangent to the GO platelets and normal to the uniaxial platelet director. In these polydomain samples, we anticipate and observe isotropic X-ray scattering patterns at normal incidence for PBDT+GO films and precursor mixtures where the beam is along the platelet director. Anisotropic, meridional and coaxial scattering patterns from edge-on measurements unambiguously demonstrate that the GO is homeotropically aligned while the polymer adopts a homogenous (polydomain) alignment, i.e., *the GO and PBDT directors are mutually orthogonal to each other*. Hence the fully nematic mixtures are locally biaxial and should be termed biaxial.

We have added the following text:

Page 6: Added 4 new references: 20,21,22,23 to manuscript.

Page 8: Added “in a single monodomain of”. The sentence now reads, “A schematic diagram of the stratified supramolecular arrangement in a single monodomain of the hN_b phase wherein the directors \mathbf{n}_{GO} and \mathbf{n}_p are orthogonal.”

Page 8: Sentence added “The random alignment of \mathbf{n}_p in the plane of the cell results in a two-dimensional mosaic structure (macroscopically uniaxial) of biaxial domains, which accounts for the isotropic scattering in the hN_b phase for $\alpha = 90^\circ$.”

2. Quantitative information on the persistence length and the overall length of PBDT must be provided.

We agree with the reviewer.

We have added the following text

Page 3: Added “with an axial persistence length of ~ 260 nm (see Supplementary Note 1) and an aspect ratio of ~ 330 .^{10,11,12}”. The sentence now reads, “This results because the rod-like PBDT double helix aggregates into prolate particles—with an axial persistence length of ~ 260 nm (see Supplementary Note 1) and an aspect ratio of ~ 330 ^{10,11,12}.”

Supplementary Information, Page 2: Added “Supplementary Note 1: Based on the I-N transition for PBDT, the effective axial rigidity persistence length (L_p) of PBDT can be calculated using Onsager theory (equation 1).^{1,2,3}

$$\phi_{\text{nematic}} \cong 4.5D/L_p \quad (1)$$

In the above equation, D denotes the PBDT rod-bundle diameter (~ 0.8 nm)² and ϕ_{nematic} is the volume fraction of rods at which the nematic phase is formed. The I-N transition occurs at 1.9 wt.%, i.e., $\phi_{\text{nematic}} = 0.015$ volume fraction assuming a bulk polymer density of 1.4 g/cm³.”

We do note that the L_p value of ~ 264 nm obtained here is a collective property and not an individual molecular property as PBDT forms supramolecular double helical bundles (Ref. 11 in text). Therefore, in the above equation, we have used the PBDT bundle diameter that has been measured previously using X-ray and NMR techniques (ref. 8 in text).

3. Due to the potential importance of electrostatic interactions in the assembly process, the authors should provide the carboxylic acid functionality of the GO, e.g., T. Szabo, E. Tombacz, E. Illes and I. Dekany, Carbon: v. 44, pp. 537–545 (2006).

We agree with the reviewer. As the reviewer suggests, GO is indeed (negatively) charged and this results from dissociated polar (oxy) functionalities (-COOH, -OH, epoxides) that are attached to the surface. A higher degree of GO functionalization will result in a higher surface charge on GO. Using XPS, we have measured the C/O ratio of our GO flakes.

We have added the following text

Page 4: Added “with a C/O ratio - indicative of functionalization degree, of 2.6 (Supplementary Fig. 3c)”. The sentence now reads, “The size selected GO platelets $\approx 3.6 \mu\text{m}$ in diameter (Supplementary Fig. 3a), consist of single GO layers according to TEM (Supplementary Fig. 3b), with a C/O ratio - indicative of functionalization degree - of 2.6 (Supplementary Fig. 3c), and in the N phase are arranged with (undulating quasi-planar) GO surfaces locally parallel.¹⁷”

Page 17: Added “**X-ray Photoelectron Spectroscopy (XPS)** measurements were performed using a Kratos Axis Ultra DLD X-ray Photoelectron Spectrometer. The measurements were performed at room temperature and at a chamber pressure of 10^{-6} mbar The GO was deposited onto Au@Si substrates prior to analysis. The binding energy reported are within ± 0.1 eV. The C/O ratio of GO was measured by dividing the atom percent of carbon by the atom percent of oxygen.”

Supplementary Information, Page 4: Figure 3c added.

4. The paper must cite and describe “Structure of DNA-cationic liposome complexes: DNA intercalation in multilamellar membranes in distinct interhelical packing regimes,”

Radler, JO; Koltover, I; Salditt, T; Safinya, CR; Science: v. 275, pp. 810-814 (1997) DOI: 10.1126/science.275.5301.810. Abstract: Cationic liposomes complexed with DNA (CL-DNA) are promising synthetically based nonviral carriers of DNA vectors for gene therapy. The solution structure of CL-DNA complexes was probed on length scales from subnanometer to micrometer by synchrotron x-ray diffraction and optical microscopy. The addition of either

linear lambda-phage or plasmid DNA to CLs resulted in an unexpected topological transition from liposomes to optically birefringent liquid-crystalline condensed globules. X-ray diffraction of the globules revealed a novel multilamellar structure with alternating lipid bilayer and DNA monolayers. The lambda-DNA chains form a one-dimensional lattice with distinct interhelical packing regimes. Remarkably, in the isoelectric point regime, the lambda-DNA interaxial spacing expands between 24.5 and 57.1 angstroms upon lipid dilution and is indicative of a long-range electrostatic-induced repulsion that is possibly enhanced by chain undulations.

We value the reviewer's suggestion.

We have added the following text:

Page 9: Sentence added “The stratified supramolecular arrangement of PBDT rods between GO platelets is reminiscent of the stratified morphology observed in a system comprising DNA rods adsorbed between 2D lipid membranes.²⁵”

5. If the authors already have examined the effects of ionic strength, counterion type and/or pH, it would enhance the present publication to mention those results (even if detailed examination of these effects is beyond the scope of this communication).

The reviewer makes a valid argument and we appreciate the suggestion. However, we are proceeding conservatively: we have only now begun to fully understand the effect of counterions (Li^+ , Ce^+ , and K^+) and ionic strength on PBDT using techniques such as rheology, scattering (X-ray/neutrons), and by tuning synthetic methods. We are currently preparing a manuscript on the effects of counterion and ionic strength on PBDT. Any study on the effects of ionic strength/counterion/pH on PBDT-GO system necessitates understanding the effects on neat PBDT.

6. In relation to strain at break, the data shows that the strain at break did not decrease (sufficient to make the point that an increase in stiffness was achieved without reducing strain at break); however, the data are not sufficient to state that the strain at break increased by 40% in PBDT+GO relative to PBDT. ED Fig 10c reveals a large standard deviation in the strain at break for the GO nanocomposites (ca. 0.34) which, combined with the small number of replicates (5), precludes a precise determination of the strain at break for the PBDT+GO nanocomposites. I was

unable to find strain-at-break values in the body or the ED; using the data in ED Fig 10c to estimate the mean strain at break (1.26% for PBDT and 1.49% for PBDT+GO) yields a difference less than 40% and a significant probability (ca. 20%) that the PBDT+GO has the same strain at break as PBDT.

We agree with the reviewer. We do note that 4 PBDT+GO samples exhibit higher strain-at-break values than the best value obtained in neat PBDT. We believe these strongly indicate an enhancement in strain. However, we also agree with the reviewer that we should not attach any particular value (such as ~40%) to this enhancement.

We have modified the text as follows:

Page 2 and Page 14: Modified “with a ~40% increase in strain-at-break” to “*without any decrease in the strain-at-break*”. The sentences now read, “The nanocomposite exceeds theoretical estimates—a 20 GPa modulus enhancement of the aramid and a strength enhancement of ~320 MPa without any decrease in the strain-at-break.” and “ We observe an unprecedented tensile strength enhancement of 320 MPa (Fig. 4b) without any decrease in the strain at break – contrary to the commonly reported propensity for brittle fracture that accompanies an increase in nanocomposite stiffness.⁴

Page 14: Sentence added “A comparison of the largest (and the average) strain-at-break values for PBDT and PBDT+GO (SI Table 3) indicates that the strain-at-break is improved in the nanocomposite films.”

Page 15: Deleted “strain-at-break”. The sentence now reads, “The striking enhancements in modulus and strength for nanocomposites prepared from hybrid biaxial nematic mixtures clearly demonstrates that rigid-rod high-performance polymers are excellent matrix materials for GO-based nanocomposites.”

Supplementary, Page 13: Added Supplementary **Table 3. Mechanical properties of PBDT and PBDT+GO nanocomposite films.** In this table we present Young’s modulus, tensile strength and strain-at-break values for PBDT and PBDT+GO films.

7. The presentation could be greatly improved. For example, the opening repeats often stated but rarely realized statement that “Reinforcing polymers with nano-scale fillers, such as carbon nanotubes or graphene oxide, is a prescription for fabricating low-density nanocomposites with exceptional mechanical properties,” which is contradicted by the cited references: ref 2 states “after nearly two decades of work in the area, questions remain about the practical impact of nanotube and graphene composites. This uncertainty stems from factors that include poor load transfer, interfacial engineering, dispersion, and viscosity-related issues that lead to processing challenges in such nanocomposites.” Indeed, *what makes this manuscript exciting is that it provides a facile, robust route to overcome processing challenges and achieve well-dispersed GO and excellent load transfer between the matrix polymer and the GO—which has proven elusive despite decades of effort.*

We agree with the reviewer and we appreciate these comments.

We have added the following text:

Page 2: Added “is often touted as”. The sentence now reads, “Reinforcing polymers with nano-scale fillers like carbon nanotubes, graphene and graphene oxide, is often touted as a prescription for fabricating low-density nanocomposites with exceptional mechanical properties.^{1,2}”

Page 2: Sentence added “In short, we present a facile, robust route to overcome processing challenges and achieve well-dispersed nanocomposites that exhibit excellent load transfer between the matrix polymer and the GO reinforcing component.”

Page 15: Added “thus enabling polymer nanocomposites with properties that have remained elusive despite decades of effort”. The sentence now reads, “Our observations suggest new design and optimization strategies for preparing nanocomposite materials from anisotropic precursors, thus enabling polymer nanocomposites with properties that have remained elusive despite decades of effort”

Reviewer #2 (Remarks to the Author):

This is an interesting paper upon the preparation and mechanical properties of a polymer-based nanocomposite reinforced by graphene oxide (GO). The authors show that considerable modulus

improvement can be realized through confining the GO in a high-modulus aligned liquid crystalline polymer, consistent with the latest theories of nanocomposite reinforcement that have been well cited in the text.

We thank reviewer 2 for these positive comments.

I have to raise a couple of issue that need further information or comment.

1) The authors show that their findings are consistent with the rule of mixtures, although as far as I can tell they only quote weight fractions of GO. Volume fractions are used in the rule of mixtures so these should be quoted, e.g. for the 7.1 wt% sample, and/or densities given so that the reader can do the conversion for themselves.

We very much appreciate the reviewer's comment, and we agree completely. We have added the densities (polymer and GO) to the Figure 4 caption. We did notice an error in the curve for theoretical estimates of ΔE values and have now fixed it in Figure 4a. The new curve does not contradict any of our statements and therefore no changes in text were necessary.

2) A number of sophisticated analytical techniques have been employed. It is notable that Raman spectroscopy, which would have been very useful for their materials (e.g. to characterize orientation or monitor stress transfer), has not been used. The authors should explain why (e.g. problems with fluorescence?) and discuss how it might have helped.

Multiple publications by Prof. R. J. Young have shown that Raman spectroscopy (Ref. 2, 3 and 4 in text) is indeed a powerful technique in assessing orientation and monitoring stress transfer between graphene and polymer. We have not attempted this yet although we are certainly exploring these studies moving forward. Given that aramids (e.g. PPTA) have been extensively investigated using Raman spectroscopy, we do not expect fluorescence to be an issue (e.g.: Knijnenberg, A., Koenders, B., Gebben, B., Klop, E., Young, R. J., van der Zwaag, S., *J. Mater. Sci.*, **45**, 2708-2714 (2010)). The analytical techniques we have employed in this paper provide reliable information on the stability of the GO colloid in a polymer solution and the orientation of both GO and polymer in both states – liquid and solid films. We agree that studying the reinforcement obtained in the composites through mechanical characterization (DMTA and stress strain) is an indirect method of validating stress transfer, and we note that we have

submitted a proposal to utilize Raman spectroscopy for studying stress-transfer in these biaxial nanocomposite films. However, the complementary techniques utilized in our manuscript unequivocally demonstrate that rigid-rod polymers are ideal hosts for GO and such composites can be easily accessed through facile mixing.

Reviewers' comments:

Reviewer #1 (Remarks to the Author):

Review of the revision of “**Strong Graphene Oxide Nanocomposites from Aqueous Hybrid Liquid Crystals**” (Co-authors include Madsen, Samulski and Dingemans) **NCOMMS-19-18812-T** Submitted via website on 7/8/19.

Due to the journal’s request that the reviewer “sign” the work when it is published, I have taken considerable time to carefully consider the rebuttal. (I will gladly prepare a drastically shortened version of the review comments if the authors would like it to be available to readers.) I continue to have the opinion that this work is worthy of publication in *Nature Communications*—provided the authors adopt wording that is compatible with the fact that they have neither evidence nor a sound argument for the hybrid nematic being biaxial (see below). Why does this detail matter? The present manuscript opens the door to a new landscape of structure-processing-property relations that will be explored by multiple research groups around the world over a sustained period of time; research over the next few years might reveal that the excellent material properties presented here are a consequence of being in the N^+ region of the phase diagram (see below). It may ultimately be found that the properties no longer improve (or possibly even suffer) when the hybrid-nematic from which the solid is obtained lies in the biaxial region of the phase diagram. (Is that the reason that the mechanical properties for the highest $C_{\text{tot}} = 14.4\text{wt}\%$ are not mentioned in this manuscript? In the marked-up article file, please search for “WHAT WAS OBSERVED FOR 14.4WT%?”) For my own peace of mind, I went through the manuscript and removed “biaxial” and subscript b wherever it was questionable: I conclude the narrative remains sound (and becomes stronger, in my opinion). I provide the “track revision” markup for the authors’ consideration.

I urge the authors to perform the following estimates for themselves and to consider the implications:

1. Please estimate the number of plates per unit volume; for 0.15 wt % GO, similar to C_{GO} in the highest C_{tot} in Figure 1c, my estimate is 8×10^{-4} plates/ μm^3 , which is less than $1/(\text{plate-excluded-volume})$. By my estimate, the plates are significantly farther apart than their lateral dimension (*contrary to the schematic Fig 2b*). This raises the distinct possibility that the normal to any one plate is uncorrelated with the normal of nearby plates (they may be GO-normals may be uniformly distributed in the plane perpendicular to the local director of the rods, in which case \mathbf{n}_{GO} is *undefined*). In the absence of compelling evidence to support a coherent director for the plates that persists over a volume that contains several plates, it is

inappropriate to express certainty that the hybrid nematic is biaxial. I urge the authors to *remove the subscript b from the labels in Figure 1c*. Any sentence that asserts that \mathbf{n}_{GO} is defined and ${}^h\mathbf{N}$ is biaxial should be worded as “*putative*” or a “*hypothesis*”.

2. Please consider how the current hybrid nematic solutions compare to the phase diagrams of Lekkerkerker 2002 [Ref 19]. To compare with Figures 2b and 4b of Ref 19, I estimated the volume fractions of rods (2×10^{-2}) and plates (5×10^{-4}); to compare with Figures 2a and 4a, I estimated the mole fraction of plates (1×10^{-8}) taking each PBDT persistence length as one rod. These are extremely low compared to the ranges covered in the figures in Ref. 19, which consider relatively low aspect ratios (rods in have $L/D=15$ and in Fig. 4, plates have $D/L=9.5$). In the present manuscript, the pure rods become fully nematic at $\phi_{rod}=0.12$ (ca. half the value for the examples in Ref 19) and the pure plates become fully nematic at $\phi_{plate}=.009$ (approximately 1/100 the value for the example in Fig 4 of Ref 19). Nevertheless, it is instructive to consider the trends predicted by the theory. The single phase biaxial nematic occurs when $\phi_{rod} > (1/2)(\phi_{rod}$ that first gives single-phase N^+ for pure rods) AND $\phi_{plate} > (1/2)(\phi_{plate}$ that first gives single-phase N^- for pure plates). For the system in the present manuscript, this would require $>6\text{wt}\%$ PBDT AND $>0.45\text{wt}\%$ GO. Neither of these is satisfied for the solutions with C_{tot} of 4.9wt% or less; and the condition for GO is not satisfied for any of the solutions considered. This is consistent with the estimate above, which showed the number concentration of GO is low compared to $1/(\text{plate-excluded-volume})$. This leads me to repeat: *I urge the authors to remove the subscript b from the labels in Figure 1c. Any sentence that implies that \mathbf{n}_{GO} is defined and/or asserts ${}^h\mathbf{N}$ is biaxial should be worded as “putative” or a “hypothesis”.*
3. Please consider the scattering pattern that would be obtained if each individual platelet adopts an orientation of its layer normal that is randomly distributed among all orientation that are orthogonal to \mathbf{n}_p . Specifically, notice that it can explain the azimuthal distribution of low q scattering observed for 4.9 and 7.1wt%. If the orientation distribution of platelets was normal to the plates of the sample cell for all of the single-phase ${}^h\mathbf{N}$, it would not be possible to have the low-q intensity more than double when concentration increases from 7.1 to 14.4wt% (as seen in Supp Fig 5b). There must be a qualitative change in the orientation distribution of platelets when the concentration increases from 7.1 to 14.4wt%—perhaps

incipient biaxial order. Search the marked-up article file for “THE PREFERENTIAL ANCHORING OF PBDT AND GO IS CONSISTENT WITH PLATELETS HAVING NORMAL ISOTROPICALLY DISTRIBUTED ABOUT \mathbf{n}_p , WHICH IS ALSO CONSISTENT WITH THE OBSERVED SCATTERING—PARTICULARLY IN VIEW OF THE NONLINEAR INCREASE OF THE STRENGTH OF THE PLATELET SCATTERING IN SUPPLEMENTARY FIG. 5b, WHICH SUGGESTS INCIPIENT BIAXIAL ORDER AT 14.4 WT%, GIVING A JUMP IN LOW Q INTENSITY RELATIVE TO 4.9 AND 7.1 WT %” and “RE: THE FOLLOWING SENTENCE, N+ HYBRID NEMATIC OF LEKKERKERKER ALSO SATISFIES \mathbf{n}_p PARALLEL TO SURFACES OF THE CELL AND PLATES”

I intersperse comments below to indicate whether or not the revised manuscript addresses the required revisions. (At the end I make recommendations regarding Figure 1 and the characterization of cited references.)

Major revisions are required prior to publication:

1. I recommend further revisions (see comments above); I appreciate the authors' rebuttal, which convinces me that hN might be biaxial at the highest concentration examined (14.4%). The present material is NOT biaxial. “biaxial” materials, including liquid crystals, have *three* distinct optical axes (e.g., cited ref 6= Mundoor, H; Park, S; Senyuk, B., Wensink, H. H., Smalyukh, I. I. “Hybrid molecular-colloidal liquid crystals.” *Science*: **360**, 768-771 (2018)). The scattering patterns observed with the x-ray beam normal to the GO are isotropic. Therefore, the present material has only *two* optical axes (all directions perpendicular to the normal to the GO are indistinguishable).
2. **Done**. Quantitative information on the persistence length and the overall length of PBDT must be provided.
3. **Done** Due to the potential importance of electrostatic interactions in the assembly process, the authors should provide the carboxylic acid functionality of the GO, e.g., T. Szabo, E. Tombacz, E. Illes and I. Dekany, *Carbon*: v. **44**, pp. 537–545 (2006).
4. **Done** The paper must cite and describe “Structure of DNA-cationic liposome complexes: DNA intercalation in multilamellar membranes in distinct interhelical packing regimes,” Radler, JO; Koltover, I; Salditt, T; Safinya, CR; *Science*: v. **275**, pp. 810-814 (1997) DOI: 10.1126/science.275.5301.810. Abstract: Cationic liposomes complexed with DNA (CL-DNA) are promising synthetically based nonviral carriers of DNA vectors for gene therapy. The solution structure of CL-DNA complexes was probed on length scales from subnanometer to micrometer by synchrotron x-ray diffraction and optical microscopy. The addition of either linear lambda-phage or plasmid DNA to CLs resulted in an unexpected topological transition from liposomes to optically birefringent liquid-

crystalline condensed globules. X-ray diffraction of the globules revealed a novel multilamellar structure with alternating lipid bilayer and DNA monolayers. The lambda-DNA chains form a one-dimensional lattice with distinct interhelical packing regimes. Remarkably, in the isoelectric point regime, the lambda-DNA interaxial spacing expands between 24.5 and 57.1 angstroms upon lipid dilution and is indicative of a long-range electrostatic-induced repulsion that is possibly enhanced by chain undulations.

5. **Response makes sense.** If the authors already have examined the effects of ionic strength, counterion type and/or pH, it would enhance the present publication to mention those results (even if detailed examination of these effects is beyond the scope of this communication).
6. **Done** In relation to strain at break, the data show that the strain at break did not decrease (sufficient to make the point that an increase in stiffness was achieved without reducing strain at break); however, the data are not sufficient to state that the strain at break increased by 40% in PBDT+GO relative to PBDT. ED Fig 10c reveals a large standard deviation in the strain at break for the GO nanocomposites (ca. 0.34) which, combined with the small number of replicates (5), precludes a precise determination of the strain at break for the PBDT+GO nanocomposites. I was unable to find strain-at-break values in the body or the ED; using the data in ED Fig 10c to estimate the mean strain at break (1.26% for PBDT and 1.49% for PBDT+GO) yields a difference less than 40% and a significant probability (ca. 20%) that the PBDT+GO has the same strain at break as PBDT.
7. **Improved** The presentation could be greatly improved. For example, the opening repeats often stated but rarely realized statement that “Reinforcing polymers with nano-scale fillers, such as carbon nanotubes or graphene oxide, is a prescription for fabricating low-density nanocomposites with exceptional mechanical properties,” which is contradicted by the cited references: ref 2 states “after nearly two decades of work in the area, questions remain about the practical impact of nanotube and graphene composites. This uncertainty stems from factors that include poor load transfer, interfacial engineering, dispersion, and viscosity-related issues that lead to processing challenges in such nanocomposites.” Indeed, what makes this manuscript exciting is that it provides a facile, robust route to overcome processing challenges and achieve well-dispersed GO and excellent load transfer between the matrix polymer and the GO—which has proven elusive despite decades of effort.

Are the images in Fig 1 (a) and (c) actual photographs? Why is the surrounding first order red? Between crossed polars, isn't the

surrounding black? My suggestion (including removal of subscript b!)→

Re: Fig 4a, look in the marked-up file for: "SHOW POLARIZER AND ANALYZER ARROWS ON IMAGES TO SIGNAL POLARIZED LIGHT WAS USED; SOLID FILM OR PRECURSOR SOLUTION? WHAT CONCENTRATIONS? 7.1wt%? 2.8wt%?"

Regarding the description of Munder *et al.*, look in the markup of the article file for: "SIGNIFICANT REVISION IS RECOMMENDED: THE BOULDER GROUP USED A SMALL MOLECULE NEMATIC SOLVENT AND DISSOLVED RODS THAT ORIENT ORTHOGONAL TO THE SOLVENT DIRECTOR---NO PLATES AT ALL!"

Look in the marked-up article file for: "PRECEDING IS ANTICIPATED BY THEORY OF LEKKERKERKER AND CO. 2002"

Reviewer #2 (Remarks to the Author):

The two reviewers both thought the paper was suitable for publication subject to revision. They each concentrated upon different aspects of the paper.

The authors have made a good response to all of the points raised and I feel the paper is now suitable for publication.

Reply to Reviewer

Strong Graphene Oxide Nanocomposites from Aqueous Hybrid Liquid Crystals

Authors: Maruti Hegde^{1,7}, Lin Yang², Francesco Vita³, Ryan J. Fox⁷, Renee van de Watering¹, Ben Norder⁴, Ugo Lafont⁵, Oriano Francescangeli³, Louis A. Madsen⁶, Stephen J. Picken⁴, Edward T. Samulski⁷, Theo J. Dingemans^{1,7*}.

*email: tjd@unc.edu

Notes: Reviewers' comments and text are shown in **red**, while author responses and new text are shown in **black**. All page numbers stated regarding the locations of modified text are from the manuscript PDF.

In the manuscript PDF, edits made based on reviewer suggestions are highlighted in **yellow**.

General Reply

In order to ensure that we are using the same nomenclature for this relatively new subject of “hybrid nematics” we preface our specific replies with a “Taxonomy of Nematics Phases” to facilitate communications.

Taxonomy of Nematic Phases

TYPES	molecular	macromolecular	colloidal
Thermotropic			
• Calamitic (rodlike)	PAA, 5CB, TBBA,...	Vectra [®] , Xydar [®] , ...	---
• Discotic	triphenylenes, ...	coal tar pitch, ...	---
Lyotropic			
• Calamitic	ionic bipyridines	PBLG, Kevlar [®] , PBDT	V ₂ O ₅ , NaYF ₄ ...
• Discotic	hydrophilic tri-phenylenes	hydrophilic coronenes	nano-clay, GO, ...

Hybrid LC = colloidal LC (lyotropic) + (macro)molecular LC (thermotropic or lyotropic)
having respective directors \mathbf{n}_c and \mathbf{n}_m

Uniaxial Hybrid Nematic = $\mathbf{n}_c \parallel \mathbf{n}_m$

Biaxial Hybrid Nematic = $\mathbf{n}_c \perp \mathbf{n}_m$

EXAMPLES OF BIAxIAL HYBRID NEMATICS:

1. *Hybrid molecular-colloidal liquid crystals* Mundoor, H., Park, S., Senyuk, B., Wensink, H. H., Smalyukh, I. I. *Science*, **360**, 768-771 (2018).

Colloidal NaYF₄ Nanorods dispersed in a thermotropic solvent 5CB. The solvent director, \mathbf{n}_{5CB} adopts normal anchoring on NaYF₄ nanorod surfaces in the mixture; the NaYF₄ nematic director is \mathbf{n}_{NaYF_4} ; the local \mathbf{n}_{5CB} directors are on average in a plane normal to the nanorod axes, hence $\mathbf{n}_{5CB} \perp \mathbf{n}_{NaYF_4} \rightarrow$ biaxial hybrid nematic, even in a quiescent (unoriented) mixture (a, b, c in figure below). A macroscopic “single hybrid liquid crystal” may be achieved by forcing the local 5CB directors to assume the same orientation on applying an external field (d). Even in this case, on a molecular length scale the 5CB long molecular axes will retain their normal anchoring at the NaYF₄ nanorod surfaces (c).

2. **Strong Graphene Oxide Nanocomposites from Aqueous Hybrid Liquid Crystals** Hegde et al *Nature Commun.* under review

Colloidal GO dispersed in a lyotropic solvent comprised of PBDT + water. The local “lyotropic solvent” director is always tangent to the GO surfaces-- \mathbf{n}_{PBDT} adopts planar anchoring on GO nanoplatelet surfaces. On a mesoscopic distance scale i.e. one where the GO director \mathbf{n}_{GO} can be defined, in a quiescent system there might be multiple domains of the PBDT phase, domains wherein there are uncorrelated orientations of \mathbf{n}_{PBDT} from domain to domain. However as in all of those domains \mathbf{n}_{PBDT} adopts planar anchoring on the (undulating) GO surface, locally $\mathbf{n}_{\text{PBDT}} \perp \mathbf{n}_{\text{GO}} \rightarrow$ a biaxial hybrid nematic phase. In the GO-PBDT mesophase, uniform ordering of the GO component is achieved because the GO colloidal liquid crystal adopts a homeotropic orientation on the casting surface: \mathbf{n}_{GO} spontaneously aligns perpendicular to the casting surface (a). In the absence of a second field (tangent to the casting surface) a “single hybrid liquid crystal” cannot be achieved. Nevertheless, in the partially aligned mixture \mathbf{n}_{PBDT} is always locally perpendicular to \mathbf{n}_{GO} .

Replies to reviewer questions:

Q 1. Due to the journal’s request that the reviewer “sign” the work when it is published, I have taken considerable time to carefully consider the rebuttal. (I will gladly prepare a drastically shortened version of the review comments if the authors would like it to be available to readers.)

Reply 1: We thank the reviewer for such a detailed review. The reviewer’s comments have undoubtedly helped make this communication better.

Q 2. I continue to have the opinion that this work is worthy of publication in Nature Communications—provided the authors adopt wording that is compatible with the fact that they have neither evidence nor a sound argument for the hybrid nematic being biaxial (see below). Why does this detail matter?

Reply 2: Our focus is indeed on the large improvements in mechanical properties when films are prepared from a nematic mixture of PBDT+GO. However, as argued below, the fluid precursor phase is a hybrid, biaxial nematic liquid crystal on a mesoscopic scale, a scale where n_{GO} can be defined (see second example above). Please recall that the hybrid biaxial nematic reported by Mundoor et al (ref. 5 in main text and 1st example above) consists of an array of quasi-parallel colloidal-size rods—a lyotropic mesoscopic uniaxial nematic—embedded in a low molar mass *thermotropic* uniaxial nematic with those respective directors orthogonal to one another even in the absence of an applied field. In the present work, we have a mesoscopic lyotropic nematic comprised of GO platelets—above the critical concentration for mesophase formation (see calculations shown in appendix below and also Reply 5)—embedded in a *lyotropic* polymeric nematic with respective directors orthogonal to one another. Apart from Lekkerkerker’s biaxial mesoscopic colloidal nematics, ours is one of the few examples of a stable rod+disk mixture, a system of persistent interest to the liquid crystal community. As emphasized below, ALL of our experimental work (phase stabilities, POM, XRD) supports the conclusion that the mixed nematic (the mesoscopic GO nematic in the lyotropic polymer nematic) develops a biaxial supramolecular organization. Does this detail (of biaxiality) matter? It may, but at the moment this is a chicken-and-egg question: Dispersions of GO platelets in amorphous polymers do not exhibit the mechanical properties of this inherently biaxial structure. So the latter organization of matrix and filler may in fact be integral to the observed physical properties.

Q 3. The present manuscript opens the door to a new landscape of structure-processing-property relations that will be explored by multiple research groups around the world over a sustained period of time; research over the next few years might reveal that the excellent material properties presented here are a consequence of being in the N^+ region of the phase diagram (see below). It may ultimately be found that the properties no longer improve (or possibly even suffer) when the hybrid-nematic from which the solid is obtained lies in the biaxial region of the phase diagram.

Is that the reason that the mechanical properties for the highest $C_{tot} = 14.4\text{wt}\%$ are not mentioned in this manuscript? In the marked-up article file, please search for “WHAT WAS OBSERVED FOR 14.4WT%?”)

Reply 3: We agree with the reviewer’s statement that this system will certainly yield rich new ideas for nanocomposite formation. We omitted the 14.4 wt.% simply because the precursor hybrid nematic is too viscous to make uniform nanocomposite films using a doctor-blade.

We have added the following text:

Page 12, added: “We do note that the prohibitively high viscosity of $C_{\text{total}}=14.4$ wt.% prevents preparation of uniform nanocomposite films using a doctor blade”.

Q 4. For my own peace of mind, I went through the manuscript and removed “biaxial” and subscript b wherever it was questionable: I conclude the narrative remains sound (and becomes stronger, in my opinion). I provide the “track revision” markup for the authors’ consideration.

Reply 4: Please also see above and replies to questions below (Replies 5-7). We can leave out the “b” subscript if the reviewer finds this notation too cumbersome; we can merely emphasize that the precursor fluid mixtures are biaxial. However, here are the facts: There is a well-defined, coherent director for the GO platelets on a mesoscopic (and on macroscopic scale in the presence of the casting surface); n_{GO} is defined by the average direction of the local normals to the undulating GO platelets. As described above, n_{GO} adopts a **homeotropic** texture, with n_{GO} normal to the casting surface. This organization in the fluid phase is indicated by the azimuthal anisotropy of the small-angle X-ray scattering in edge-on measurements (observe the scattering close to the beamstop in SI fig. 5a, also shown in Reply 7), and by the highly reduced low-angle, isotropic scattering in the corresponding normal incidence measurements. At the same time, the n_{P} director adopts a **planar** texture with n_{P} tangent to the casting surface (and GO platelet surfaces). Based on the observed angular dependent X-ray diffraction, in **both the precursor hybrid mixtures** and in the dried **films**, n_{P} is tangent to the macroscopically-aligned GO surfaces that, in turn, have their normals (n_{GO}) perpendicular to the casting surface and the dried film surface. So, in the PBDT+GO precursor mixture with the PBDT nematic pervading the exfoliated GO nematic, the n_{P} and n_{GO} directors are normal to one another. The only option for this arrangement of well-defined directors in the liquid crystalline precursor fluid mixture is a **biaxial** arrangement of plates and rods wherein the directors n_{P} and n_{GO} are well defined.

Q 5. Please estimate the number of plates per unit volume; for 0.15 wt % GO, similar to CGO in the highest C_{tot} in Figure 1c, my estimate is 8×10^{-4} plates/ μm^3 , which is less than $1/(\text{plate-excluded-volume})$. By my estimate, the plates are significantly farther apart than their lateral dimension (contrary to the schematic Fig 2b). This raises the distinct possibility that the normal to any one plate is uncorrelated with the normal of nearby plates (they may be GO-normals may be uniformly distributed in the plane perpendicular to the local director of the rods, in which case n_{GO} is undefined). In the absence of compelling evidence to support a coherent director for the plates that persists over a volume that contains several plates, it is inappropriate to express certainty that the hybrid nematic is biaxial. I urge the authors to remove the subscript b from the labels in Figure 1c. Any sentence that asserts that n_{GO} is defined and hN is biaxial should be worded as “putative” or a “hypothesis”.

Reply 5: We thank the referee for calling attention to the GO number density and distance between GO plates at the concentrations used throughout our *Communication*. Based on the reviewer’s questions, we have explicitly calculated the interplatelet distances in our mixtures (in the SI and) incorporated clarifying changes in the manuscript summarized at the end of this reply.

Before discussing the calculated interplatelet distances, we reiterate the experimental evidence supporting our assertion that n_{GO} is well defined and that the hybrid nematic is biaxial.

1. We clearly see that the precursor mixtures are nematic; there is no phase-separated isotropic solution above $C_{total} = 3.6$ wt.% (figure 1).
2. If GO is dispersed in a nematic PBDT “solvent,” but is below the critical phase transition for forming a GO nematic (for, e.g, $C_{total} = 1.7$ wt.% and in mixtures of PBDT+*small GO*, fig. 4, S.I.), the GO will phase separate forming an isotropic layer rapidly on centrifuging or more slowly on standing. This is similar to observations made by Lekkerkerker’s work on rod+disk mixtures (ref. 19). In contrast, we do NOT observe phase separation into N^- and N^+ phases for the mixtures as observed by Lekkerkerker, even on a time scale of 12 months.
3. The hybrid mixture behaves differently from the pure components: For $C_{total} = 4.75$ wt.% the hybrid mixture exhibits a stable hN_b phase, whereas the individual component solutions are biphasic (see table 1, S.I. and discussion on page 7 in manuscript).

The interesting proposal by the referee—an isotropic distribution of non-liquid crystalline GO platelets having their normal perpendicular to n_P —is precluded by 1 and 2 above, and experimental observations amplified in Reply 7.

As suggested by the reviewer, it *does* make sense to compare our results to the predictions of Lekkerkerker's model for hybrid systems. However, one should also consider that: **i)** Onsager's model does not quantitatively describe the transition concentrations of pure GO solutions— $\Phi_{I-N} = 2.1 \frac{t}{r} = 9.3 \times 10^{-4}$ and $\Phi_N = 2.83 \frac{t}{r} = 1.2 \times 10^{-3}$ do not match experimental values, $\Phi_{I-N} = 9.0 \times 10^{-5}$ and $\Phi_N = 4.5 \times 10^{-3}$. **ii)** Lekkerkerker's 2002 paper (ref. 19) considers rods and disks with aspect ratios (and hence excluded volumes) very different from those of our system. **iii)** Lekkerkerker himself recognizes that, "the effect of polydispersity and the influence of higher-order particle correlations (both are not incorporated here) may give rise to qualitatively different scenarios from the ones predicted by our calculations.”

Calculations: We have included a calculation of the inter-GO platelet distances and critical overlap concentration in the revised manuscript (SI Table 2 and associated calculations; see Appendix below). We find the referee’s estimate of the number density of GO plates at ~0.15 wt % GO in the hybrid precursor mixture too low by two orders of magnitude. In our nematic mixtures, the calculated inter-GO platelet distances are in a regime where excluded volume interactions between the GO platelets dominate. That is, GO exceeds its critical overlap concentration Φ_{GO}^* and forms a colloidal lyotropic nematic phase. This occurs in neat GO solutions when $\Phi_{GO} > 0.018$ wt.%, and in the precursor

mixtures at $C_{\text{total}} \geq 2.0$ wt.%; see SI Table 2 or Table in Appendix below. As a result, the normals to the basal planes of the GO platelets *are* orientationally correlated and the scheme depicted in Figure 2b is qualitatively correct.

We have added the following text:

Page 6, added: In nematic hybrid mixtures i.e. when $C_{\text{total}} > 2.0$ wt.%, GO exceeds its critical overlap concentration (ϕ_{GO}^*) (see Supplementary Table 2 and associated calculations in S.I.) resulting in orientational correlation between GO platelet normals. Furthermore, the effective volume per GO platelet ($V_{\text{eff,GO}}$) which is a measure of the accessible volume for GO platelets reduces below the corresponding GO overlap volume (Supplementary Table 2).

Supplementary Information, page 5, added: We have added a table (Table 2, titled: Parameters relevant to hybrid nematic phase formation in PBDT+GO precursor mixtures) along with the calculation methodology (shown here in Appendix below).

Q 6. Please consider how the current hybrid nematic solutions compare to the phase diagrams of Lekkerkerker 2002 [Ref 19]. To compare with Figures 2b and 4b of Ref 19, I estimated the volume fractions of rods (2×10^{-2}) and plates (5×10^{-4}); to compare with Figures 2a and 4a, I estimated the mole fraction of plates (1×10^{-8}) taking each PBDT persistence length as one rod. These are extremely low compared to the ranges covered in the figures in Ref. 19, which consider relatively low aspect ratios (rods in have $L/D=15$ and in Fig. 4, plates have $D/L=9.5$). In the present manuscript, the pure rods become fully nematic at $\phi_{\text{rod}}=0.12$ (ca. half the value for the examples in Ref 19) and the pure plates become fully nematic at $\phi_{\text{plate}}=.009$ (approximately 1/100 the value for the example in Fig 4 of Ref 19). Nevertheless, it is instructive to consider the trends predicted by the theory. The single phase biaxial nematic occurs when $\phi_{\text{rod}} > (1/2)(\phi_{\text{rod}}$ that first gives single-phase N+ for pure rods) AND $\phi_{\text{plate}} > (1/2)(\phi_{\text{plate}}$ that first gives single-phase N- for pure plates). For the system in the present manuscript, this would require $>6\text{wt}\%$ PBDT AND $>0.45\text{wt}\%$ GO. Neither of these is satisfied for the solutions with C_{tot} of 4.9wt% or less; and the condition for GO is not satisfied for any of the solutions considered. This is consistent with the estimate above, which showed the number concentration of GO is low compared to $1/(\text{plate-excluded- volume})$. This leads me to repeat: I urge the authors to remove the subscript b from the labels in Figure 1c. Any sentence that implies that nGO is defined and/or asserts hN is biaxial should be worded as “putative” or a “hypothesis”.

Reply 6: The error in the referee’s calculation of the number density of GO platelets and the corresponding inter-GO platelet distance makes most of the remarks in Q 6 moot. Regarding the comparison between our results and Lekkerkerker’s model, see comments in our reply to Q5. Additionally, Onsager and others (refs: Eppenga, D. Frenkel, *Mol. Phys.* **1984**, *52*, 1303 and M. A. Bates, D. Frenkel, *J. Chem. Phys.* **1999**, *110*, 6553) have noted that theoretical predictions for phase transitions for platelets are less accurate than for rods, (ref: R. Tkacz et al, *Part. Part. Syst. Charact.* **2017**, *34*, 1600391). Also see reply Q7 below.

Q 7. Please consider the scattering pattern that would be obtained if each individual platelet adopts an orientation of its layer normal that is randomly distributed among all orientation that are orthogonal to nP. Specifically, notice that it can explain the azimuthal distribution of low q scattering observed for 4.9 and 7.1wt%. If the orientation distribution

of platelets was normal to the plates of the sample cell for all of the single-phase hN , it would not be possible to have the low q intensity more than double when concentration increases from 7.1 to 14.4 wt% (as seen in Supp Fig 5b). There must be a qualitative change in the orientation distribution of platelets when the concentration increases from 7.1 to 14.4 wt%—perhaps incipient biaxial order. Search the marked-up article file for “THE PREFERENTIAL ANCHORING OF PBDT AND GO IS CONSISTENT WITH PLATELETS HAVING NORMAL ISOTROPICALLY DISTRIBUTED ABOUT n_P , WHICH IS ALSO CONSISTENT WITH THE OBSERVED SCATTERING— PARTICULARLY IN VIEW OF THE NONLINEAR INCREASE OF THE STRENGTH OF THE PLATELET SCATTERING IN SUPPLEMENTARY FIG. 5b, WHICH SUGGESTS INCIPIENT BIAxIAL ORDER AT 14.4 WT%, GIVING A JUMP IN LOW Q INTENSITY RELATIVE TO 4.9 AND 7.1 WT %” and “RE: THE FOLLOWING SENTENCE, N+ HYBRID NEMATIC OF LEKKERKERKER ALSO SATISFIES n_P PARALLEL TO SURFACES OF THE CELL AND PLATES”

Reply 7:

Our calculations (Reply 5 and Table in Appendix) indicate that GO platelet normals are orientationally correlated at concentrations above $C_{total} = 2.0$ wt.%, i.e., when GO is mixed with the nematic PBDT phase with director n_P (defined by the quasi-parallel PBDT chain contours), GO exceeds its critical overlap concentration for solutions (see Appendix below) and GO forms a colloidal mesophase. And as a result, both the anisotropic scattering in edge-on SAXS measurements and the isotropic low q scattering in normal measurements are inconsistent with the model suggested by the reviewer, i.e., an isotropic distribution of GO platelet normals around the PBDT director n_P . The SAXS data is also inconsistent with a model wherein the GO directors are isotropically distributed in a plane normal to n_P .

Additionally, the scattering intensities for different C_{total} values in Supplementary Fig. 5b are reported in arbitrary units and are not directly comparable—measurements were taken on different samples with minor experimental variations (sample thickness, shear stresses, alignment relative to the X-ray beam etc). Moreover, the arbitrary intensity vs. q plot is also constructed from a radial cut of the SAXS scattering patterns i.e. the radial vector lies along the GO and PBDT scattering) obtained at $\alpha \approx 15^\circ$ for all mixtures. The increase in small-angle scattering with the concentration is therefore not surprising and can be attributed to i) the increase in the GO concentration and ii) the consequent increase of the GO nematic order parameter.

When GO platelets are oriented with their surface normal orthogonal to the casting surface i.e. they are homeotropically aligned, the resulting low q scattering at normal incidence ($\alpha = 90^\circ$) should have an isotropic azimuthal profile i.e. circular (for example, see Figure 2 in Malwitz, M. M. et al, Orientation of Platelets in Multilayered Nanocomposite Polymer Films. *J. Polym. Sci., Part B: Polym. Phys.*, **41**, 3237-3248 (2003)). Importantly, the homeotropic GO platelet organization should result in anisotropic low q scattering in edge-on measurements ($\alpha = 15^\circ$) (Figure 2 in Malwitz, M. M. et al in paper cited above).

In the biaxial precursor mixtures, the observed low q scattering has an isotropic azimuthal profile at normal incidence ($\alpha = 90^\circ$). The observation in the edge-on ($\alpha \approx 15^\circ$) measurements of a low q anisotropic scattering corresponds to GO platelets strongly confined to the plane of the casting mixture—no out-of-plane distribution relative to the cell plane normal. This is well evident at concentrations $C_{\text{total}} \geq 4.9$ wt% (SI fig. 5a, also shown here for convenience). This unequivocally confirms that the platelets are confined to the plane of the precursor casting mixture (a homeotropic organization with GO platelets tangent to the casting surface) and are quenched in the plane of the resulting dried films (see SAXS of nanocomposite films, Supplementary Fig. 7a). The polydomain in-plane orientation of n_p results in isotropic azimuthal scattering profile at larger q for normal incidence ($\alpha = 90^\circ$) and coaxial azimuthal intensity variation for edge-on incidence ($\alpha = 15^\circ$) (Fig. 2a and Supplementary Figure 5a). The angular dependence of the scattering patterns along with the coaxial, anisotropic scattering of PBDT and GO at ($\alpha = 15^\circ$) requires the biaxial arrangement in the fluid (hybrid) phase— n_p tangent to n_{GO} in the fluid precursor solutions.

In contrast, the model proposed by the reviewer (GO-platelet normals randomly distributed orthogonal to n_p), one would expect a low q scattering distribution with an isotropic azimuthal profile for both normal incidence and edge-on measurements. In addition, the reviewer suggests that the hybrid mixture is entering the biaxial phase only at the highest concentration ($C_{\text{total}} = 14.4$ wt%): in that case, one would expect a reduction of low q scattering at normal incidence, which is contrary to our observations.

In summary, we conclude that the precursor hybrid mixture PBDT+GO mixture is biaxial, not just by looking at the azimuthal intensity variation at $\alpha = 15^\circ$ but also by considering the absence of scattering at $\alpha = 90^\circ$ (summarized in table below).

MODEL	X-RAY ALIGNMENT		
		NORMAL INCIDENCE ($\alpha = 90^\circ$)	EDGE-ON ($\alpha = 15^\circ$)
random planar N^+ phase (reviewer's model)	low q	isotropic scattering	isotropic scattering
	larger q	isotropic ring	2 azimuthal peaks
N_b phase	low q	isotropic scattering	azimuthal intensity variation
	larger q	isotropic ring	2 azimuthal peaks, coaxial with the low q scattering

Our observations match the expected results for N_b phase. Since our biaxial hybrid phase is formed by mixing a colloidal nematic with a molecular nematic, we have designated this as a hN_b phase.

Changes made to the text:

Page 8-9, modified: “ The following sentence, “ In the edge-on geometry, the small-angle GO-dominated scattering transforms from a circular pattern ~~in the isotropic and the biphasic mixtures~~, to an anisotropic azimuthal intensity distribution in the fully nematic mixtures (Fig. 2a and Supplementary Fig. 5a) was modified to,

“In the edge-on geometry, the small-angle (low q) GO-dominated scattering transforms from a circular pattern to an anisotropic azimuthal intensity distribution in the fully nematic mixtures (Fig. 2a and Supplementary Fig. 5a).”

Page 9, added: “...(Fig. 2a and Supplementary Fig. 5b)...”

Supplementary Information, page 8, added: “The scattering intensities for different C_{total} values in Supplementary Fig. 5b are reported in arbitrary units and are not directly comparable—measurements were performed on different samples with minor experimental variations.”

Supplementary Information, page 8, Fig. 5b, modified: We have now removed the numbers associated with the Y-axis as these are arbitrary numbers.

(Above figure is Figure 5a in SI and shown here for convenience)

Q 8. Are the images in Fig 1 (a) and (c) actual photographs? Why is the surrounding first order red? Between crossed polars, isn't the surrounding black?

Reply 8: The images in Fig 1 (a) are actual polarized optical microscopy images. To place the images into context, we have cropped the rectangular POM images into test-tubes shapes using Adobe Illustrator.

The images in Fig. 1c are actual digital photographs between crossed polarizers. We have artificially changed the background color to enhance the contrast.

We have added the following text:

Page 4, added: “...images from crossed polarized optical microscopy have been cropped into NMR tube shapes using Adobe Illustrator™)”

Page 6, added: “The backgrounds in Fig. 1a, Fig. 1c and Fig. 1d have been changed from black to pink to aid visualization of the hybrid precursor mixtures, especially the isotropic fractions.”

Q 9. Re: Fig 4a, look in the marked-up file for: “SHOW POLARIZER AND ANALYZER ARROWS ON IMAGES TO SIGNAL POLARIZED LIGHT WAS USED; SOLID FILM OR PRECURSOR SOLUTION? WHAT CONCENTRATIONS? 7.1wt%? 2.8wt%?”

Reply 9: The optical images are of nanocomposite **films** obtained **without polarizers**. These optical images, together with those in SI fig. 10a, demonstrate that significant aggregation occurs when isotropic or biphasic hybrid solutions ($C_{\text{tot}} = 1.7$ or 2.8%) are used for nanocomposite film preparation. The figure caption does state that the top image

corresponds to **films** obtained from $C_{\text{tot}} = 7.1\%$ precursor mixture and the bottom image corresponds to films obtained from $C_{\text{tot}} = 2.8\%$.

Q 10. Regarding the description of Mundoor et al., look in the markup of the article file for: “SIGNIFICANT REVISION IS RECOMMENDED: THE BOULDER GROUP USED A SMALL MOLECULE NEMATIC SOLVENT AND DISSOLVED RODS THAT ORIENT ORTHOGONAL TO THE SOLVENT DIRECTOR---NO PLATES AT ALL!”

Reply 10: In the manuscript, the sentence that follows the Mundoor et al. citation states, “In the latter, mesoscopic **rodlike particles** were added to a **thermotropic molecular nematic**, which adopted normal anchoring relative to the particle surfaces resulting in two orthogonal nematic directors.” Orthogonal directors are the signature of a biaxial phase. Mixing a mesoscopic or colloidal nematic with a molecular nematic defines a hybrid phase and that is the reason we have cited their paper and adopt their terminology-- **a hybrid nematic**.

We have added the following text:

Page 2, added, “...—a mesoscopic lyotropic nematic comprised of GO platelets embedded in a lyotropic polymeric nematic with respective directors orthogonal—”

Q 11. In Supplementary Information, the reviewer inserted the following text (shown here in **bold+italics**) on page 3, “The size selected GO platelets $\approx 3.6 \mu\text{m}$ in diameter (Supplementary Fig. 3a), consist of [**NOT ACCORDING TO THE SUPPLEMENTARY INFO**] single GO layers according to TEM (Supplementary Fig. 3b), with a C/O ratio - indicative of functionalization degree - of 2.6 (Supplementary Fig. 3c), and in the N^+ phase are arranged with (undulating quasi-planar) GO surfaces locally parallel.¹⁷”

Reply 11: Our TEM experiments have shown that most GO platelets are bilayer or monolayer (Supplementary Figure 3).

We have modified the sentence to reflect this statement:

Page 3, modified: “...and bilayer...”

Q12. The reviewer inserted the following text below Supplementary Table 1, “It might help the readers if the definitions of the superscripts + and – are reiterated in this caption.

Reply 12: We thank the reviewer for the suggestion. We have defined all the abbreviations below the table to aid the reader.

We have added the following text:

Supplementary Information, page 8, added: “Abbreviations: C_{total} = total solids concentration = (PBDT+GO)/(PBDT+GO+H₂O); expressed in wt.%, C_P = concentration of polymer in hybrid mixture, C_{GO} = concentration of GO in hybrid mixture, N^+ = nematic phase formed by PBDT rods, N^- = mesoscopic nematic phase formed by GO platelets, I = isotropic, hN_b = hybrid biaxial nematic”

Q 13. The reviewer inserted the following text (shown here in **bold+italics**) on page 7, “***DESPITE LONG STANDING THEORETICAL PREDICTIONS [CITE],*** we are unaware of prior reports of stable rod+plate mesophases as de-mixing occurs spontaneously.²⁴”

Reply 13: We thank the reviewer for this addition. We have accepted the change and cited references accordingly.

The sentence now reads:

Page 7, added: “Despite long standing theoretical predictions,^{19,21} ...”

Additional: We found a typo in the experimental section.

We have modified the following text:

Page 20, deleted: “A 10 mm gauge length was used for both stress-strain and dynamic mechanical thermal analysis (DMTA) measurements.”

APPENDIX: CALCULATION of NUMBER DENSITY and INTERPLATELET DISTANCE

The volume of a single GO platelet = $V_{GO} = \pi r^2 t = 0.00815 \mu\text{m}^3$ (here r = radius of GO platelet = $1.8 \mu\text{m}$, t = thickness = 0.8 nm). The expression $\frac{1}{d} = \sum \frac{C_i}{d_i}$ yields the density (d) of the PBBDT+GO mixture; d_{GO} is $\sim 2.0 \text{ g/cm}^3$ (ref. 6 in S.I) and C_i and d_i are mass concentrations (wt. %) and densities of individual components (GO, PBBDT and water). At $C_{GO} = 0.15 \text{ wt.}\%$ (used by the reviewer), $C_P = 6.0 \text{ wt.}\%$ (to maintain mass fraction $F_{GO} = 0.0244$), and $d \sim 1.02 \text{ g/cm}^3$. The volume fraction of graphene oxide (ϕ_{GO}) in the mixture in turn can be calculated using $\phi_{GO} = \frac{d}{d_{GO}} C_{GO}$. The volume fraction $\phi_{GO} = 0.00076$ for $C_{GO} = 0.15 \text{ wt.}\%$ in the mixture. The number density of GO platelets $\rho_{NGO} = \frac{\phi_{GO}}{V_{GO}} = 0.094 \mu\text{m}^{-3}$. This is two orders of magnitude greater than the reviewer’s estimate ($8 \times 10^{-4} \mu\text{m}^{-3}$). The effective volume per GO platelet (V_{eff}), i.e., the accessible volume per GO platelet in the mixture, can be calculated using $V_{\text{eff}} = \frac{1}{\rho_{NGO}} = 10.66 \mu\text{m}^3$ per GO platelet. According to Onsager, the excluded volume, $V_{\text{excl, GO}}$ of randomly oriented GO platelets with thickness t and diameter D , is given by:

$$V_{\text{excl, GO}} = \frac{1}{4} \pi D \left(L^2 + \frac{1}{2} (\pi + 3) t D + \frac{1}{4} t D^2 \right)$$

Since $t \ll D$, this can be approximated to $V_{\text{excl}} = \frac{\pi^2}{16} D^3 \sim 29 \mu\text{m}^3$ (D = diameter of GO = $3.6 \mu\text{m}$). We can make a rough estimate the inter-platelet distance using the expression: $H = \frac{t}{\phi_{GO}} = 1.1 \mu\text{m}$. **Both V_{eff} and H are lower than**

$V_{\text{excl, GO}}$ and D , indicating significant overlap of GO platelets.

Additionally, the critical overlap concentration for GO (ϕ_{GO}^*) can also be calculated using the following expression:

$$\phi_{GO}^* = \frac{V_{\text{disk}}}{V_{\text{swept sphere}}} = \frac{3t}{4r} = 0.00033,$$

Supplementary Table 2: Parameters relevant to oriented phase formation for PBDT-GO systems.

C_{total}	C_P	C_{GO}	Density of precursor mixture (d)	ϕ_{GO}	V_{eff}	H
(wt.%)	(wt.%)	(wt.%)	g/cm ³		μm ³	μm
1.7 (I)	1.67	0.041	1.005	0.00021	39.56	3.9
2.8 (I+ ^h N _b)	2.75	0.069	1.008	0.00035	23.43	2.3
4.9 (^h N _b)	4.75	0.12	1.014	0.00061	13.39	1.3
7.1 (^h N _b)	6.89	0.173	1.021	0.00088	9.23	0.9
14.4 (^h N _b)	14.04	0.351	1.044	0.00183	4.45	0.4

C_{total} = total solids concentration = (PBDT+GO)/(PBDT+GO+H₂O); expressed in wt.%, C_P = concentration of polymer in hybrid mixture, C_{GO} = concentration of GO in hybrid mixture, N⁺ = nematic phase formed by PBDT rods, N⁻ = mesoscopic nematic phase formed by GO platelets, I = isotropic, ^hN_b = hybrid biaxial nematic. ϕ_{GO} = volume fraction of graphene oxide, V_{eff} = effective volume per GO platelet, H = interplatelet distance.

In the table above, we have listed the V_{eff} for all the hybrid precursor mixtures.

By coincidence, *the biphasic mixture ($C_{tot} = 2.8\%$) exhibits V_{eff} and ϕ_{GO} that is close to $V_{excl,GO}$ and ϕ_{GO}^* .*

REVIEWERS' COMMENTS:

Reviewer #1 (Remarks to the Author):

The additional information regarding the size, shape and mass of the GO particles is very helpful. Two missing details are needed:

- a) The scale bar in SI Fig 3a (left) is missing; I had thought it was the same as the scale bar in SI Fig 3a (right), until I realized that the "flake size" histogram extends beyond 8microns in contradiction to the longest span across the largest "flake" in SI Fig 3a (left) being less than 8-times the length of the 1micron scale bar. Similarly, estimating the effective diameter as $4(\text{Area})/(\text{Perimeter})$ yields a value less than the nominal average of 3.6microns even for the largest particle shown if the 1micron scale bar is used.
- b) The definition of "flake size" is missing in the caption of SI Fig 3a. Was it the longest span? Or $4(\text{Area})/(\text{Perimeter})$? Or something else? Averaged over how many particles?

I look forward to seeing this communication in print and seeing the research that it inspires.

--Julia A. Kornfield

Reply to Reviewer

Strong Graphene Oxide Nanocomposites from Aqueous Hybrid Liquid Crystals

Authors: Maruti Hegde^{1,2}, Lin Yang³, Francesco Vita⁴, Ryan J. Fox¹, Renee van de Watering¹, Ben Norder⁵, Ugo Lafont⁶, Oriano Francescangeli⁴, Louis A. Madsen⁷, Stephen J. Picken⁵, Edward T. Samulski¹, Theo J. Dingemans^{1,2*}

*email: tjd@unc.edu

Notes: Reviewers' comments and text are shown in **red**, while author responses and new text are shown in **black**. All page numbers stated regarding the locations of modified text are from the Supporting Information file. In accordance with editorial formatting requirements, no highlights have been made in the Supporting Information file.

Reviewer #1:

The additional information regarding the size, shape and mass of the GO particles is very helpful. Two missing details are needed:

a) The scale bar in SI Fig 3a (left) is missing; I had thought it was the same as the scale bar in SI Fig 3a (right), until I realized that the “flake size” histogram extends beyond 8microns in contradiction to the longest span across the largest “flake” in SI Fig 3a (left) being less than 8-times the length of the 1micron scale bar. Similarly, estimating the effective diameter as $4(\text{Area})/(\text{Perimeter})$ yields a value less than the nominal average of 3.6microns even for the largest particle shown if the 1micron scale bar is used.

Response: The scale bars for the two images are indeed **not** the same. We thank the reviewer for spotting this missing information. The scale bar (10 μm) has now been added to the figure (Left image of Supplementary Figure 3a).

b) The definition of “flake size” is missing in the caption of SI Fig 3a. Was it the longest span? Or $4(\text{Area})/(\text{Perimeter})$? Or something else? Averaged over how many particles?

Response: Flake size and particle count number have now been defined in text. We have added the following text (page 4) to the figure legend (shown in **bold+italics**), “*Statistical GO flake size analysis of ~120 GO flakes* yields an average lateral flake size of 3.64 μm with a polydispersity of 63%. *Flake size is defined as the longest span and was measured using ImageJ software.*”

I look forward to seeing this communication in print and seeing the research that it inspires.

We thank the referee for her very detailed review, which has significantly improved the quality of our manuscript.